

**Spatial and Temporal Variation of Bulk Snow Properties in**
**North Boreal and Tundra Environments Based on**
**Extensive Field Measurements**
**H.-R. Hannula[1], J. Lemmetyinen[1], A. Kontu[1], C. Derksen[2], and J. Pulliainen[1]**
[1]{Arctic Research, Finnish Meteorological Institute, Sodankylä, Finland}
[2]{Climate Research Division, Environment Canada, Toronto, Ontario, Canada}
Correspondence to: H.-R. Hannula (henna-reetta.hannula@fmi.fi)



**Abstract**
In this paper, an extensive dataset of snow *in situ* measurements, collected in support of
airborne SAR-acquisitions in Sodankylä and Saariselkä test sites in northern Finland, is used
to analyse the heterogeneity of bulk snow properties (snow depth, density and water
equivalent) over different land cover types in northern taiga and tundra areas. In addition, the
applicability of different spatial frequencies of snow sampling to estimate the true snow
conditions is investigated. Overall, the highest variability in bulk snow properties was found
over sparsely vegetated land cover groups, but the scale of variation was smaller in forested
areas, as these areas exhibited a low correlation length in snow depth. This implies that more
frequent measurements should be executed in forested (~ every < 5 m) than in open areas (~
every 7.5-12.5 m) to catch the true variability in snow depth. The results also indicated that
the current spatial resolutions of space borne microwave radiometers and radars used for the
remote retrieval of bulk snow properties are all well above the limit to fully describe the
spatial variation of e.g. snow depth even in open areas. This conclusion supports the demand
of research investigating high-resolution parameter retrieval in remote sensing of snow, e.g.
using advanced SAR techniques.





## 1    Introduction

Snow is a temporally and spatially variable component of global climate and hydrological
systems. Due to its high reflectivity, insulation and water holding capacity, changes in snow
cover are crucial, for example, in water supply forecasts, and in ecological, climatic, and
meteorological studies (Vaughan et al., 2013). Accurate snow information is needed to succesfully
parametrisize the snow component in numerical weather prediction (NWP) models (de
Rosnay et al., 2014). The on- and offset of the annual snow cover is also linked to the carbon
balance in the northern latitudes via soil freezing and respiration processes (Grogan and
Jonasson, 2006).
Several studies have reported explicit changes in snow cover extent (SCE) and snow cover
duration (SCD) over both Northern Hemisphere and Arctic snow cover in recent decades (e.g.
Brown and Mote, 2009; Brown and Robinson, 2011; Choi et al., 2010). Many of these studies
show explicit decreasing trends in the snow cover extent (e.g. Brown et al., 2010; Derksen
and Brown, 2012) but with differing responses during the winter and spring months and in
North America and in Eurasia (Brown, 2000).
Snow properties have been succesfully measured via satellites over several decades. Although
remote sensing (RS) methodologies themselves have changed little, the available imagery is
very different. Significant improvements have been achieved in the resolution of the retrieved
information (Nolin, 2010) and in our understanding of the snow cover as a part of the global
climate system. However, as even in flat areas snow properties vary in a number of different
scales (Derksen et al., 2010; Sturm and Benson, 2004), snow information retrieval via satellite
RS remains challenging. Typically one has to trade off spatial resolution for better temporal
resolution, and vice versa, while vegetation, topography, and later, simplified snow physics in
retrieval algorithms introduce error into the end products (Foster et al., 2005). These
uncertainties can significantly affect our understanding of the current snow cover changes and
could lead to biased evaluation of global climate models and erroneous input to NWP models
(Frei and Lee, 2010).
The lack of extensive ground data collected simultaneously with the RS data is often a
limitation for further assessment and development of RS algorithms. Furthermore, sufficient
resolution to describe the variation of different snow parameters is not always clear. At what
scale can we measure accurately enough to catch the relevant snow characteristics and what,
in general, is the effect of the scale on the interpreted output information? At the moment, the



best resolution achieved in the observation of snow parameters by optical sensors is in order
of 250-500 m (Hall et al., 2002; Notarnicola et al., 2013), whereas passive microwave
radiometers are limited to resolutions of tens of kilometres. Using synthetic aperture radar
(SAR), resolutions ranging from, for example, one meter (e.g. TerraSAR-X/TanDEM-X) up
to 50 km (ENVISAT/ASAR) can be achieved (Dietz et al., 2013). However, the revisit times
of space-borne radars are, even at best, typically limited to several days at high latitudes,
whereas passive instruments, at the expense of resolution, can have daily coverage over a
large part of the Northern Hemisphere (Dietz et al., 2013). Understanding the signal response
within a 625 $km^2$ footprint produced by radiometers is challenging (Foster et al., 2005; Nolin,
2010). Microwave methodologies are generally practical for monitoring SWE and SD,
whereas optical instruments are suitable for SCE and albedo measurements. The optical
methods suffer from the lack of sufficient sun-light and frequent cloud cover in the northern
latitudes (Warren, 1982).
In this paper, an extensive field measurement dataset acquired in Finland is used to quantify
the temporal and spatial heterogeneity of different bulk snow parameters in the northern taiga
and tundra environments. A large collection of *in situ* snow data was collected in support of
ESA SnowSAR airborne acquisitions in Northern Finland during the winter of 2011-2012
(Lemmetyinen et al., 2014). During the measurement campaign, the mission concept of the
proposed ESA CoReH2O (Cold Regions Hydrology High-resolution Observatory, ESA 2012;
Rott et al., 2010) mission, at that time a candidate for the ESA Earth Explorer-7 satellite, was
demonstrated. The main objectives of this study are 1) to characterize the temporal and spatial
heterogeneity of bulk snow properties by calculation and comparison of land cover specific
statistics of snow depth (SD), snow water equivalent (SWE), and snow density, 2) to
investigate the temporal changes in snow stratigraphy over different types of land cover by
description of snow stratigraphy changes, and 3) give an estimate of an optimal sample
frequency for SD measurements by autocorrelation analysis and investigate the applicability
of different sampling frequencies to estimate the true snow conditions.



## 2   Data and methods

### 2.1   Study sites

The *in situ* data used in this study were collected in support of ESA SnowSAR airborne acquisitions, which occurred over three sites during the winter of 2011-2012. Most acquisitions were located at the primary site, an approximately 7 by 10 km area close to the FMI Arctic Research Centre (FMI-ARC) located in Sodankylä northern Finland. Each airborne mission attempted to cover the entire area using a mosaic of up to 30 flight transects. Acquisitions were timed to correspond closely to the planned CoreH2O revisit times during the two proposed phases of the mission (3 and 15-day revisit time). The main site represents a typical boreal forest/taiga environment dominated by spruce/scots pine forests of varying density, as well as open peatbogs (wetlands) (Fig. 1 left). The elevation in the area varies between 180 m and 240 m above sea level and is relatively flat. The area covered by the acquisitions also included several rivers and lakes.

The second site was situated ~150 km north of the primary site in Saariselkä region (Fig. 1 right), representing an upland tundra environment. The area is mainly treeless, but the ground vegetation is characterized by lichen, mosses, sprigs and some larger shrubs, which result in a more varying distribution of snow cover due to wind effects. The general topography was also more variable with several low-lying tundra hills situated along the acquisition path. This site was visited twice during the season; a single ~20 km transect was covered. The aim was to provide data for CoReH2O retrieval performance testing over the tundra land cover type, which was not well represented at the main site. A third site of acquisitions was located over sea ice in the Gulf of Bothnia, but these data are not covered here.

### 2.2   Data collection

The airborne acquisitions were aimed to follow a 15-day repeat period between December and mid-February, corresponding to the repeat-pass time of CoReH2O during the second phase of the mission. For a period between February 22 and March 9, a three-day repeat period was planned, corresponding to the planned CoReH2O repeat pass time during the first phase of the mission (See dates in Table 1). A total of ten airborne acquisitions, as well as one dedicated calibration mission, were flown at the main site. Two acquisitions were flown at the Saariselkä tundra site. Ground sampling at the main site took place on most occasions during



the day of the airborne acquisitions. On occasion, sampling was continued on the day
following a flight, if the snow conditions remained stable. In Saariselkä, ground data was
collected only on the dates of the airborne acquisitions.
Manual sampling of SD and SWE along flight transects formed the core of the *in situ* data
collection in support of each SnowSAR acquisition. Figure 1 shows the locations of the
collected ground measurements at the Sodankylä and the Saariselkä test sites. The basic
concept was to, at minimum, cover at least two 5 km transects for each flight. Snow depth
was sampled every 100 m while SWE was sampled every 500 m. As a goal, the sampling of
designated tracks was designed to take place within 200 meters (across-track) from the
planned centrelines of the flight transects. Sampling teams moved either on foot (snowshoes),
by skis, or by snowmobiles. At each sampling site, snow depth was recorded at minimum
from three representative locations in a 10 m radius, while the SWE measurement was taken
from one representative location. An automated geolocated snow depth measuring tool
('Magnaprobe') was also used on all sampling days. For transects where the Magnaprobe was
employed, SD measurements were considerably more frequent in distance (approximately
every 2-10 meters).
On individual tracks, the measurements were conducted at approximately the same locations
for every SnowSAR mission, to minimize the disturbance of the snowpack in the
measurement area. The main objective of the distributed measurements was to obtain a
maximal amount of SD/SWE samples for comparison with the airborne observations. Around
600 SWE, 22 100 SD measurements were collected during a total of 19 days between
December 2011 and March 2012 (Table 1). Additionally, the manual snow measurement
program of FMI-ARC provided snow pit observations at three sites in the Sodankylä area,
enabling to construct a time series of the physical evolution of snow over dry mineral soil,
wetlands and lake ice during the campaign (Leppänen et al., 2015).

### 2.3   Data analysis


#### 2.3.1   Data processing and analysis of snow heterogeneity


Erroneous data points (e.g. snow depths smaller than 1 cm) and duplicates were removed
from the dataset. Based on GPS coordinates, for each measurement point, land cover class
was determined. The land cover information was available through the European Commission
programme to COoRdinate INformation on the Environment (Corine). An updated dataset



(CLC2012) was used. Analogously to former airborne data analysis from the site
(Lemmetyinen et al., 2015) the original 44 CLC2012 land cover classes were generalized into
nine land cover groups (Table 2). The spatial coverages of different land cover groups within
a 7 km x 10 km area in the both test sites, used later in the analysis, are also shown. Forested
areas were divided based on both the tree canopy closure (>30 % dense/ <30% sparse) and the
soil type (mineral/peat or organic). Different types of open areas were also separated
(wetlands, meadows, barren surfaces, water systems). The ninth group included all artificial
surfaces, such as roads and buildings, and were excluded from the analysis.
The division of the measurements based on canopy closure as well as the overall land cover
class is justifiable because canopy closure has been observed to be one of the main factors to
affect snow accumulation (e.g. Dobre et al., 2012; Storck et al., 2002). This applies especially
to flat areas, such as the Sodankylä region, where elevation and aspect have little effect
(D'Eon 2004). In addition, RS of snow cover has proven to be problematic in forested regions
(e.g. Foster et al., 2005; Heinilä et al., 2014). In the boreal forest zone, the vegetation itself
has large effect on the RS measurements and needs to be taken into account (Cohen et al.,
2015; Derksen, 2008; Metsämäki et al., 2012) whereas in tundra regions the high proportion
of frozen lakes, local scale variability due to wind effects, and stratigraphically complicated
snowpack introduces different kinds of problems (Derksen et al., 2010).
Density information for each snow depth measurement point was determined based on the
distributed SWE measurements. Since fewer SWE than SD measurements were available, the
density was calculated per day per land cover group. If more than one SWE points were
measured within the same land cover group during the same day, an average of these
measurements was used. In case no density information for a distinct land cover group was
available, data from the previous or the subsequent measurement day was used, if no
precipitation events or drastic temperature changes had occurred. The variation of air
temperature and the daily precipitation amount at FMI-ARC in Sodankylä during the
measurement campaign are shown in Fig. 2. After the density determination, SWE for each
SD data point was calculated. For some data points no density and thus SWE information
could be determined. The number of SD and SWE measurements within each generalized
land cover group during each measurement day is represented in Table 1. Finally, a boxplot
for each land cover group for each measurement day was created to describe the temporal
variation in the snow properties during the measurement campaign.


The evolution of snow physical properties (grain size, density, stratigraphy, and temperature)
was analysed from snow pit information collected from three sites (sparse forest on dry
mineral soil (SFm), wetland (OB), and lake ice (LR)).

### 2.3.2   Autocorrelation of snow depth measurements

The statistical variability of SD over different types of land cover was investigated by means
of analysing the autocorrelation of measured SD values over distance. Snow depths measured
with the Magnaprobe instrument were applied, as these provided the necessary high spatial
sampling frequency. The goal was to estimate the optimal sampling frequency for snow cover
in different land cover conditions, informing future planning of snow sampling campaigns in
the region, and to identify deficiencies of the relatively sparse sampling approach applied
elsewhere during the campaign (SD every 100 meters, SWE every 500 meters).
In order to harmonize the analysis, multiple transects of 500 meters were chosen from the
collected data, representing each investigated land cover group. Autocorrelation was
calculated as a function of lag distance. An exponential fit was applied to the autocorrelation,
deriving the exponential (auto) correlation length ($L_{ex}$). However, the data did not cover all
land cover groups for all SnowSAR acquisitions with a sufficient amount of samples to
conduct the autocorrelation analysis. The autocorrelation analysis was applied only for SD as
SWE was estimated for each SD measurement point via land cover type fixed density and
would have produced same results as the previous analysis.

### 2.3.3   Effect of sampling frequency

To further investigate the effect of sampling frequency, the average SD obtained via the
frequently executed Magnaprobe measurements and, the more sparsely executed
measurements with a snow ruler (henceforth called conventional SD measurements), were
statistically compared. The goal was to assess if the different sampling frequencies
(Magnaprobe, potential over-sampling / conventional, potential under-sampling) lead to a
statistically significant difference in the mean SD. Three land cover groups (DFm/OB/LR),
characterized by different average SD, and measurement days comprising a sufficient amount
of both Magnaprobe and conventional measurements were chosen for the comparison.
For the analysis, each sub-group of the measurements was tested for normality by histograms
and by the Kolmogorov-Smirnov test for later selection of appropriate statistical analysis. For



part of the groups the assumption of normality did not hold. The equality of variances
between the groups was tested as well by executing both the Levene's and the Bartlett's tests.
If the test results were inconsistent, the histograms were investigated to assess, which result
could hold better. As the assumption of equal variances also did not hold between all the
compared groups, and the sample sizes varied, finally, both the Welsch's t-test for unequal
variances (assumes normality but not equal variances) as well as the Mann-Whitney U-test
(MWU) (assumes equal variances but not normality) were chosen to test the statistical
difference of means.
Furthermore, as *in situ* data is often averaged over the field of view of RS observations for
validation purposes, it was investigated, if different measurement frequencies lead into
different outcomes when a weighted average, based on land cover proportions within a typical
RS observation grid cell, is calculated. For this purpose a 7 km x 10 km area was cut out from
the generalized CLC2012 land cover data and percentual coverages of each generalized land
cover group, both in the Sodankylä and in the Saariselkä test sites, were determined (Table 2).
This area was approximately equivalent to the spatial extent of the ground measurements in
the Sodankylä test site. In the Saariselkä test site, the ground sampling occurred on a slightly
smaller area, but for the comparison purposes, areas of same sizes were chosen. Four
campaign days with a high amount of measurements were chosen for the analysis; three days
from the Sodankylä test site and one day from the Saariselkä test site, as enough
measurements were available from Saariselkä only from the second acquisition day. The
distances between all the consequent measurements were calculated and three different cases
of measurement frequencies were compiled; one with maximum sampling frequency (~ every
1-10 m), one with medium sampling frequency (~ every 100 m), and one with sparse
sampling frequency (~ every 500 m). However, as the measurement distances varied and were
not always exactly e.g. every 100 m, it was not possible to produce withholded data with the
exact sampling frequencies; the sampling frequencies, for example, in the 100 m case may
actually vary between ~70-150 m. However, the sampling frequencies of the three cases were
still clearly different. A proportionally weighted average for both SD and SWE were
calculated separately for each case of measurement frequency.


## 3   Results

### 3.1   Land cover specific variation of snow properties

The median SD of the lakes and rivers was distinctly lower than those of the other surface type groups during the whole measurement campaign (Fig. 3). The deviation of snow depth in the lakes and rivers was also generally lower (or during few days, as high as) the deviation in the other land cover groups. An exception occurs in the beginning of March when comparatively high deviation in the measured SD was seen. One possible explanation for this is that in the beginning of March, most of the measurements were taken on river ice whereas during the other days the measurements were mainly on lake ice. The narrow creeks might have larger SD variation than open lakes. Another possible reason is the imprecision in retrieving the CLC2012 data for the measurement points. The resolution of the CLC2012 is 20 m and the handheld GPS devices used during the ground data collection may have inaccuracies of several meters. This could have led to an incorrect classification of, for example, adjacent forest measurements as lake and river measurements in the narrow creek areas.

Another distinctive group was the open bogs, also with lower median SD values. The difference between the open bogs and the other groups was more significant in the beginning and in the end of the measurement campaign than in the middle. In the forest groups, the median SD of the dense forests on mineral soil was lower than the median SD of the dense forests on peat soil during most of the campaign period. One possible explanation for this could be a different canopy structure in the forests on mineral and peat/organic soils. The relationship between the sparse forests on mineral and peat soils was similar, but the differences in the median SD were smaller and the relationship was not as clear as between the dense forest groups. The median SD of the fields and meadows and the barren land were not clearly lower or higher than that of the forest groups (DFm/DFp/SFm/SFp). Overall the median SD of the fields and meadows and the barren land cover groups was slightly lower than the median SD in the forest cover groups and a bit higher than that on the open bogs. The highest deviation in SD was measured during the Saariselkä measurement day (29Feb). This indicates well the effect of elevation changes and wind on the SD variation on open tundra site in relation to the taiga forest site in Sodankylä. Regarding the median and deviation of the SWE measurements in the different land cover groups (Fig. 4), very much similar results than





with the SD values were obtained, but the differences, for example, between the two dense
and sparse forest groups were slightly easier to detect.
For the density calculations fewer measurements were available and on some days a single
measurement might represent the snow density value in a land cover group (Fig. 5). The
median density was highest in the field and meadows and the barren lands measured in the
Saariselkä test site. This is explained by the wind effect which packs the snow and easily
introduces larger densities than in the taiga test site. The effect of wind and elevation changes
were also seen in the high deviation in the measurements made in February 29th. The
deviation of snow density on the lakes and rivers was very large when looking both the
median density values and the difference between the minimum and the maximum values; on
some days, the density of this group was lower, and on some days much higher than in any
other land cover group. The wind also played a role in the open water areas but probably even
more, the high density variation was related to water which might rise to the ice surface on
mild air temperatures. The comparatively thin snow layer also thaws easily during warm days
in spring. For the density values, the relationship between the values of the land cover groups
was different from those of SD and SWE, as densities were typically higher in the open areas
than in the forested areas. The relationship between the different forested land cover groups
was not clear, the dense forests on mineral soil exhibited lower densities than the other
forested land cover groups. It was also notable that the variation in density between the
different land cover groups did not stay constant but as environmental factors changed, snow
bulk density in the different land cover groups reacted differently; on some days all the
measured densities of the land cover groups were very close to each other, and on other days
large differences existed.
The evolution of snow stratigraphy over dry mineral soil, wetlands and lake ice during the
campaign are depicted in Fig. 6. Snow structure evolved from December to March by addition
of new fine-grained snow layers on the surface (deep blue in the Fig. 6) and grain growth in
the lower half of the snowpack (from light blue through green and yellow to orange). These
effects were also visible in the snow density profiles: new snow on the surface was very light,
and the density of the bottom layers increased throughout the winter. Snow structure in the
forest on mineral soil and over the open bog was very similar, at least compared to snow on
the lake ice; similar layers could be detected from the forest and the open bog profiles, even
though from different heights. Typically snow depth on the bog was smaller than in the forest,



but the heavy snowfall in February (between 8 and 22 Feb) evened out the difference.
Temperature profiles reflect the fact that air temperature was the same at all pits measured on
the same day; the differences in the snow surface temperatures can be explained by the
differing measurement times. The disparity between the temperature profiles in e.g. 9 Jan is
due to differing snow depths.
**3.2   Analysis of snow depth autocorrelation**
The autocorrelation of the measured SD values over distance was analysed to statistically
describe the variability of SD. Examples of the autocorrelation of the measured SD over
representative transects are shown in Fig. 7 for the forested areas (DFm) and the wetlands
(OB). An exponential fit to the autocorrelation is shown. The correlation length (in meters)
derived from the fit is also displayed, providing a measure of the degree of variability in snow
depth over distance. The forested sample exhibited the lowest $L_{ex}$, while autocorrelation
remained high over longer distances over the wetland transect.
The mean and standard deviation of $L_{ex}$, derived for the different land cover groups, is
summarized in Table 3. The values were calculated from representative 500 m transects
selected from all Magnaprobe sampling campaigns at the main site, as well as the second
Saariselkä campaign.  The barren land cover type represents data collected from the second
Saariselkä campaign (average and standard deviation of $L_{ex}$ from 21 transects), while only one
suitable transect was available from the main site representing the sparse forests on peat soil.
On average, the forested areas exhibited a low correlation length in snow depth, while values
collected over lake ice and wetlands exhibited correlation lengths in excess of 15 meters. Over
the barren landscape in Saariselkä, the average autocorrelation was in excess of 20 meters.
This can be explained by the influence of the forest canopy, which affected the spatial
distribution of snow accumulation on the forest floor, inducing a large variability over short
distances. However, when the mean coefficient of variation (CV) for each land cover group,
representing the whole campaign period, was calculated (Table 4), the dispersion of SD was
highest on the lakes and rivers and on the barren lands and remained low in the forested land
cover groups and on the fields and meadows.



### 3.3 Effect of sampling frequencies

The results from the statistical analysis of the difference of means in SD are represented in Table 5. Both of the statistical tests gave similar results when the means of frequent Magnaprobe measurements and the sparse conventional measurements were compared. Only on 26Feb in the dense forests on mineral soil the Welch's test estimated the difference to be significant whereas MWU-test estimated it insignificant (at a 0.05 level of confidence). During most of the days, difference of SD means was significant in the dense forests on mineral soil. This supports the result of the autocorrelation analysis, where snow depth varied in short distances due to e.g. forest canopy effects. Two days with measurements from the lakes and rivers were analysed and during both days the difference in SD means was statistically significant. The results from the comparisons of the open bogs were not consistent; during the first two days compared, the results were not statistically significant but during rest of the days, they were.

The weighted averages of SD and SWE for the 7 km x 10 km areas are presented in Table 6. A consistent effect of the measurement frequency could not be determined; during the first two days investigated, the weighted averages increased slightly as the sampling frequency was decreased. However, during the second last day investigated (23Feb), the weighted SD and SWE values decreased slightly between the most frequent and the 100m sampling case. The 500m sampling case did not introduce change in these average values. In the Saariselkä test site (29Feb), the differences between the three cases of sampling frequency were larger and were first decreasing and then increasing along the sparser measurement frequency. The last two dates had the most frequently measured snow parameters and the measurement frequency could be most accurately manipulated. Overall, the averaged SWE values changed more along with the sampling frequency than the values of SD. As SWE values were retrieved by using the land cover specific density values, an additional source of error in the frequent SWE dataset existed. For any robust conclusion, a more comprehensive analysis of the effects of the sampling frequency needs to be done with a dataset, in which frequency changes are optimised for this kind of study.



## 4    Discussion


There are three often-mentioned generalizations about the spatial variability of snow.
According to the first one, seasonal snowpacks are more heterogeneous than perennial
snowpacks due to higher amount of acting agents (Sturm and Benson, 2004). In addition to
wind and water percolation effects, vegetation and topographic changes affect seasonal
snowpacks whereas in perennial snow, the first two factors are the most important. Secondly,
it is generally thought that snow is less spatially variable in forested than in open areas (in
arctic and subarctic), as in open areas the wind redistribution effectively increases the
heterogeneity (Derksen et al., 2014; Essery and Pomeroy, 2004). Thirdly, slightly
contradictory to the second generalization, variation of SD and SWE is often thought to be
higher in forested areas, where complicated canopy structure affects the snow accumulation
on the ground, than, for example, in open areas (Dobre et al., 2012; Storck et al., 2002). In
this study, the  heterogeneity of bulk snow properties (SD, density and SWE) was analysed
from three different perspectives; by statistical description of bulk snow properties in different
land cover groups, by autocorrelation analysis (for SD only), and by the determination of an
averaged CV for each land cover group investigated.
According to the statistical description of bulk snow properties (e.g. Fig. 3 and 4), deviations
in snow depth between the land cover groups were often small, being in the order of 1-3 cm.
In addition, the relative differences did not remain constant (e.g. deviation in the forested
areas was not always higher than in the open areas), but varied even during the mid-winter.
Only the bulk snow properties measured at the Saariselkä test site had consistently higher
deviations in the measured values of SD, density and SWE than the values at the main test site
in Sodankylä. This supports the earlier results of the variation of snow properties over tundra
(e.g. Derksen et al., 2014). However, the autocorrelation analysis presented in the Sect. 3.1.2,
revealed that the snow properties tended to vary more on short distances in forests, although,
on average the deviation in forests was no higher than over sparsely vegetated land cover
groups. This supports the third generalization mentioned in the previous section. Lastly, the
mean values of CV revealed an opposite phenomenon as the lowest CV values were observed
in the forested land cover groups (Table 4). This implies that although the absolute variation
in snow depth was largest in the sparsely vegetated groups, this heterogeneity appeared on
scales larger than that in the forested groups. By analysing the autocorrelation of measured
SD values over distance in different land cover groups, the proper sampling frequency



capturing the true variation in the measured quantity, can be determined. The $L_{ex}$ averages for the different land cover groups shown in Table 3 indicated that in open areas a measurement frequency of 7.5-12.5 m for SD was adequate. Ideally, the sampling should be executed at least twice more frequently than $L_{ex}$ so that the true variance could be captured. In forested areas, the snow depth should be measured every < 5 m to catch the true variation of SD. These results, however, show that the sampling frequency used usually in the conventional snow course measurements (e.g. SD every 50 m, SWE every 500 m for Finnish Environment Institute snow course measurements) is not optimal for the full spatial description of SD. Explicit improvement in the data quality could be achieved by following the presented measurement guidelines for the spatial sampling frequency. With respect to RS applications, instruments measuring at resolutions higher than the land cover group specific $L_{ex}$ do not provide meaningful statistical information compared to instruments whose resolutions are close to the values of $L_{ex}$.

The need for higher sampling frequency in the forested areas was supported by the analysis presented in the Sect. 3.1.3, where with only one exception, the difference in mean SD obtained via the Magnaprobe and the conventional measurements, was statistically significant in the dense forests on mineral soil. On the lakes and rivers the differences were also statistically significant during the both days investigated. This could be related to the overall high variation of SD on lakes and rivers (Table 4). On the open bogs, the differences became statistically significant towards the end of the campaign with increasing snow depth. It is hypothesized that this is because the overall deviation in SD tends to increase along with snow depth, and as such, different sampling frequencies might have more effect.

The variation of all bulk snow parameters at the Saariselkä test site was very high. This indicates that retrieval of snow parameters in a tundra region with only mild elevation changes (highest fells were usually around 500 m), and where the vegetation effect is almost non-existent, can still be very complex. Furthermore, factors such as aspect and elevation, which were not considered in this study, should be taken into account. The effect of different sampling frequencies on the spatially averaged values of SD and SWE was not clear (Table 6). The analysis gives some robust references that the effect might be more significant in SWE than in SD measurements and that the effect increased as the SD increased along the snow season. One could also hypothesize that the sampling frequency might affect more in the tundra site, as the bulk snow property variation in tundra is generally very high and the





obtained differences in this analysis were larger than the ones obtained in the Sodankylä test
site. However, a more sophisticated analysis needs to be done to properly evaluate the effect
of different sampling frequencies on the spatially averaged bulk snow property values, which
are often used for RS data validation purposes.
The temporal changes in snow stratigraphy in the different types of land cover (Fig. 6)
revealed that the evolution was rather similar between the sparse forest on mineral soil and on
the open bog; the main differences were seen in the bottom of the snowpack where the local
microtopography can significantly increase the spatial and temporal heterogeneity of the
snowpacks (Sturm and Benson, 2004). The snowpack stratigraphy on the lake ice was very
different from the other two snowpacks with lower SD and fewer snow layers.

## 5  Conclusions

In this study an extensive dataset of *in situ* snow measurement, collected in support of ESA
SnowSAR airborne acquisitions in Northern Finland during the winter of 2011-2012, was
used to describe the temporal and spatial heterogeneity of bulk snow properties in different
types of land cover in tundra and taiga test sites. The optimal sampling frequency for SD
measurements was investigated by means of an autocorrelation analysis of measurements
with a high sampling rate. The applicability of different sampling frequencies to estimate the
true snow conditions was analysed. This can also be useful to inform the development of RS
methodologies, as the spatial and temporal heterogeneity of snow is one of the main
challenges for correct RS information retrieval.
The results revealed that although, on average, the deviation of bulk snow properties in the
forest land cover groups was not higher than in the open land cover groups, snow properties
tended to vary more over short distances in forests. On the other hand, the absolute variance,
described by the averaged coefficient of variation for each land cover group, showed the
highest dispersion of the measurement values in the open land cover groups. This indicates
that although the absolute variance in forests was lower than in the other groups, more
frequent sampling procedure should be applied to fully catch the bulk snow property variation
in the forested types of land cover. A measurement frequency of 7.5-12.5 m is adequate in
open bogs and lakes and rivers. In forested areas the snow depth should be measured around
every < 5 m to catch the true variation of SD.



With respect to remote sensing applications, the results showed that the current spatial
resolutions of the space borne radiometers and radars used for remote SD retrievals are all
well above the limit to fully describe the spatial variation of snow depth even in open areas.
The conclusion supports the demand of research investigating high-resolution parameter
retrieval in RS of snow, e.g. using advanced SAR techniques.
In the future work this extensive snow ground dataset will be further analysed and utilized
together with simultaneously observed airborne and space-borne RS observations, with the
goal of developing novel retrieval algorithms for snow geophysical properties.
**Acknowledgements**
The work was supported by the EU 7th Framework Program project "European-Russian
Centre for cooperation in the Arctic and Sub-Arctic environmental and climate research"
(EuRuCAS, grant 295068), by the A4-project (Arctic Absorbing Aerosols and Albedo of
Snow, decision No. 254195), Nordic Top-level Research Initiative (TRI) "Cryosphere-
atmosphere interactions in a changing Arctic climate" (CRAICC), funded by the Academy of
Finland, and by Maj and Tor Nessling Foundation.



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

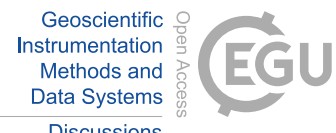

Table 1: The *in situ* measurement dates, the number of snow depth and SWE/density measurements, and the snow pit observations
(* + Mar13) in the different generalized land cover groups. Dates marked in **bold** indicate measurements days conducted in the
Saariselkä test site.

| Land cover group SD SWE | Dec 19 | **Dec 20** | Jan 9 | Jan 10 | Jan 23 | Jan 24 | Feb 7 | Feb 8 | Feb 9 | Feb 22 | Feb 23 | Feb 24 | Feb 25 | Feb 26 | **Feb 29** | Mar 1 | Mar 5 | Mar 8 | Mar 23 | Total |
|---|---|---|---|---|---|---|---|---|---|---|---|---|---|---|---|---|---|---|---|---|
| DFm | 91 | x | 195 | x | 172 | 218 | 86 | 20 | 17 | 587 | 1018 | 177 | 658 | 116 | 11 | 1238 | 22 | 58 | 23 | 4887 |
|  | 4 | x | 4 | x | 7 | 8 | 7 | 6 | 3 | 13 | 13 | 1 | 1 | 12 | x | 16 | 7 | 6 | 5 | 113 |
| DFp | 30 | x | 25 | x | 24 | 72 | 23 | 23 | 5 | 193 | 209 | 98 | 58 | 55 | x | 134 | 7 | 41 | 11 | 1008 |
|  | 3 | x | 5 | x | 2 | 4 | 1 | 2 | 2 | 4 | 4 | x | x | 2 | x | 1 | 2 | 1 | 3 | 36 |
| SFm* | 95* | x | 3* | x | 74 | 81 | 22* | 45* | 13 | 241* | 126 | 53 | 44 | 32 | 16 | 179 | 15 | 36* | 13* | 1120 |
|  | 4 | x | 2 | x | 4 | 2 | 1 | 2 | 4 | 6 | 2 | x | x | 4 | x | 6 | 2 | 4 | 3 | 46 |
| SFp | 19 | x | 46 | x | 17 | 38 | 34 | 11 | 3 | 141 | 92 | 78 | 73 | 63 | x | 201 | 8 | 53 | 7 | 884 |
|  | 1 | x | 2 | x | 2 | 2 | 3 | 1 | x | 4 | 1 | 2 | 1 | 5 | x | 4 | 5 | 3 | 1 | 37 |
| FM | 17 | 11 | 35 | x | 37 | 7 | 37 | 28 | x | 138 | 146 | 7 | 10 | 45 | 3154 | 111 | x | x | 5 | 3788 |
|  | x | 3 | 1 | x | 2 | 2 | 1 | 4 | x | 10 | x | 1 | 1 | 1 | 60 | 3 | x | x | x | 89 |
| B | x | 5 | x | x | 2 | x | x | 2 | x | x | x | x | x | x | 588 | x | x | x | x | 597 |
|  | x | 1 | x | x | x | x | x | x | x | x | x | x | x | x | 11 | x | x | x | x | 12 |
| OB* | 331* | x | 333* | 262 | 123 | 69 | 281 | 55* | 45 | 598* | 407 | 791 | 2761 | 380 | x | 1052 | 60 | 300* | 92* | 7940 |
|  | 14 | x | 25 | 7 | 23 | 5 | 11 | 5 | 9 | 27 | 8 | 8 | 13 | 23 | x | 23 | 9 | 18 | 20 | 248 |
| LR* | x* | x | 42* | x | 375 | 3 | 32 | 254* | 1* | 1098* | 1 | x | x | x | x | 6 | 4 | 10 | 57* | 1883 |
|  | x | x | 1 | x | 10 | x | 2 | 10 | 1 | 8 | 1 | x | x | x | x | 1 | 1 | 2 | 4 | 41 |





Table 2: Generalization of the field measurements based on the CLC2012 land cover classes
analogously to Lemmetyinen et al. (2015), and the spatial coverage (%) of each land cover group
within a 7 km x 10 km area in the Sodankylä and the Saariselkä test sites.

| Acronym for generalized land cover group | Description | CLC2012 classes | 7x10 km area coverage (%) | |
|---|---|---|---|---|
| | | | Sodankylä | Saariselkä |
| DFm | Dense forests (mineral soil type) | 22,24,26,27,29 | 33.37 | 30.80 |
| DFp | Dense forests (organic/peat soil type) | 23,25,28 | 10.06 | 0.72 |
| SFm | Sparse forests (mineral soil type) | 33,35,36 | 7.85 | 5.32 |
| SFp | Sparse forests (organic/peat soil type) | 34 | 6.08 | 0.28 |
| FM | Fields and meadows | 16,17,18,19,20,21,30,31,32 | 4.84 | 50.40 |
| B | Barren | 37,38,39 | 0.02 | 10.94 |
| OB | Open Bogs | 40,41,42,44,45 | 25.68 | 1.22 |
| LR | Lakes and rivers | 46,47,(48) | 10.81 | 0.06 |
| O | Other (roads and urban areas) | 1-15,43 | 1.29 | 0.25 |





Table 3: The mean and standard deviation of exponential autocorrelation length ($L_{ex}$) of snow depth
over the land cover groups, calculated from representative transects during the snow sampling
campaigns between Dec 19, 2011 and March 23, 2012.

| | land cover class | DFp | DFm | SFp | SFm | OB | FM | B | LR |
|---|---|---|---|---|---|---|---|---|---|
| $L_{ex}$ | mean | 6.8 | 5.6 | 4.0 | 1.5 | 15.5 | 9.2 | 21.1 | 15.8 |
| | stdev | 3.6 | 1.9 | - | 0.7 | 7.8 | 6.2 | 15.0 | 5.3 |





Table 4. The averaged coefficient of variation for snow depth within each land cover group during
the measurement campaign.

| Land cover group | Coefficient of variation SD |
|---|---|
| DFm | 0.16 |
| DFp | 0.13 |
| SFm | 0.13 |
| SFp | 0.13 |
| FM | 0.17 |
| B | 0.33 |
| OB | 0.20 |
| LR | 0.36 |



Table 5: The statistical difference of means between the Magnaprobe and the conventional
measurements within the dense forests on mineral soil, the open bogs, and the lakes and rivers. The
p-value is marked with * if the result is statistically significant at a significance level of 0.05. The
test type considered more appropriate for each individual group is marked in **bold**.

| Date | Land cover class | n | | Mean | | Welch's t-test | | | Mann-Whitney U | | |
|---|---|---|---|---|---|---|---|---|---|---|---|
| | | Magna | conv | Magna | conv | Df | t-statistic | p | Df | MWU-statistic | p |
| 19Dec | DFm | 71 | 20 | 28.43 | 32.30 | 29.16 | 2.596 | 0.014* | 89 | 445.5 | **0.011*** |
| | OB | 233 | 98 | 23.00 | 22.99 | 225.34 | -0.021 | **0.983** | 329 | 9993.0 | 0.073 |
| 23Jan | DFm | 151 | 21 | 46.37 | 50.30 | 25.55 | 2.547 | 0.017* | 170 | 1000.0 | **0.006*** |
| | LR | 305 | 70 | 18.52 | 21.72 | 88.19 | 3.031 | **0.003*** | 373 | 8224.0 | 0.003* |
| Jan24 | DFm | 191 | 27 | 44.84 | 47.97 | 34.04 | 2.188 | 0.036* | 216 | 1841.5 | **0.016*** |
| | OB | 46 | 23 | 46.21 | 41.70 | 34.97 | -2.068 | 0.046 | 67 | 372.0 | **0.046** |
| 7Feb | DFm | 54 | 32 | 48.54 | 53.84 | 68.85 | 3.730 | 0.000* | 84 | 457.0 | **0.000*** |
| | OB | 209 | 72 | 48.23 | 45.22 | 105.87 | -2.506 | 0.014* | 279 | 5563.0 | **0.001*** |
| 8Feb | DFm | 129 | 71 | 45.62 | 51.64 | 179.14 | 5.524 | **0.000*** | 198 | 2545.5 | 0.000* |
| | LR | 173 | 81 | 19.65 | 22.14 | 153.93 | 4.559 | **0.000*** | 252 | 4561.0 | **0.000*** |
| 22Feb | DFm | 550 | 37 | 69.48 | 70.71 | 48.99 | 1.065 | **0.292** | 585 | 9742.5 | 0.665 |
| | OB | 446 | 152 | 66.10 | 57.38 | 291.35 | -8.591 | **0.000*** | 596 | 19659.5 | 0.000* |
| 26Feb | DFM | 94 | 22 | 65.21 | 69.44 | 53.18 | 2.284 | **0.026*** | 114 | 828.0 | **0.148** |
| | OB | 308 | 72 | 57.38 | 50.86 | 120.08 | -5.317 | **0.000*** | 378 | 6883.0 | **0.000*** |



Table 6: The proportionally weighted averages (WA) of SD and SWE for the 7 km x 10 km land
areas in the Sodankylä and the *Saariselkä* test sites for the three different cases of measurement
frequency.

| | SD | | | SWE | | |
|---|---|---|---|---|---|---|
| | Magnaprobe WA | 100 m WA | 500 m WA | Magnaprobe WA | 100 m WA | 500 m WA |
| 19Dec | 25.07 | 25.05 | 25.35 | 50.13 | 52.62 | 52.89 |
| 23Jan | 38.50 | 40.48 | 43.01 | 67.40 | 76.41 | 84.06 |
| 23Feb | 68.82 | 66.65 | 66.62 | 133.74 | 129.73 | 129.73 |
| *29Feb* | 50.06 | 40.74 | 47.45 | 135.91 | 110.33 | 129.43 |



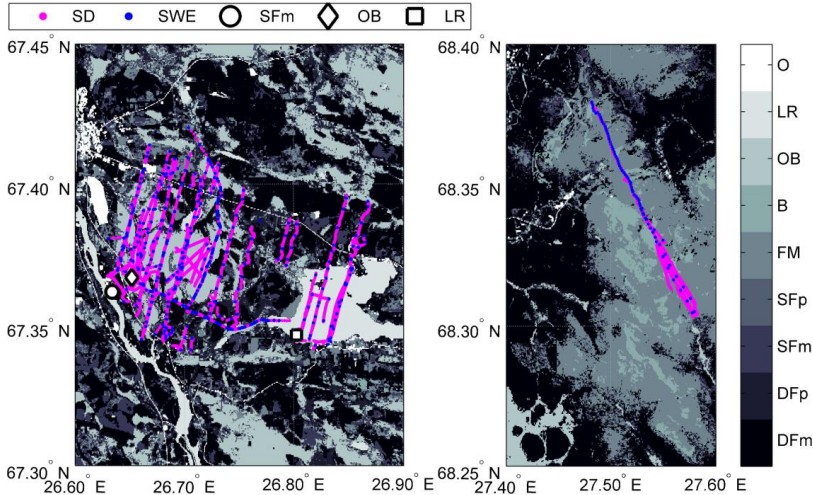


Figure 1: Snow depth, SWE, and snowpit measurements (three different sites) collected in the
Sodankylä (left) and the Saariselkä (right) test sites during the SnowSAR acquisitions.





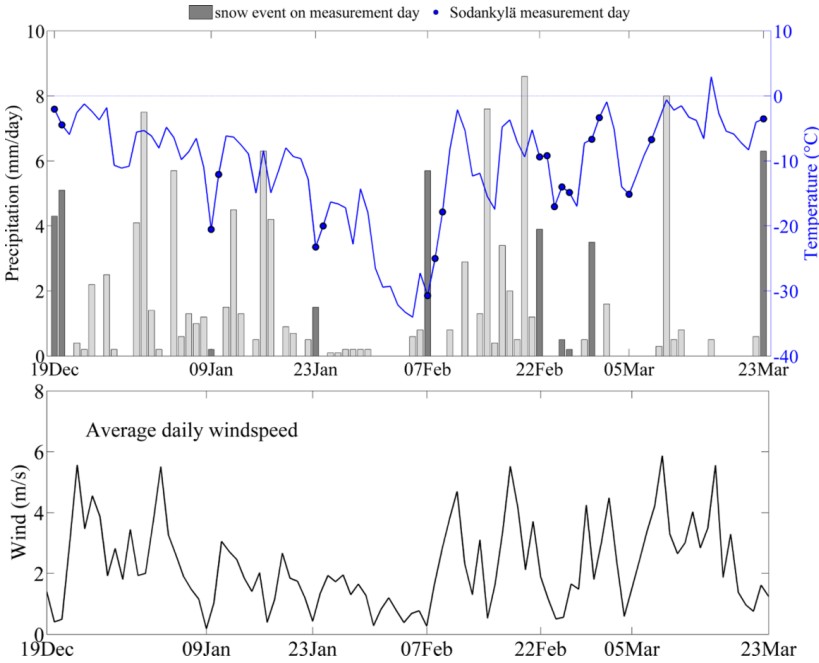

Figure 2: Daily average temperature, daily precipitation sum, and average daily wind speed during
and between the different field measurement days observed by the automatic weather station at
FMI-ARC.





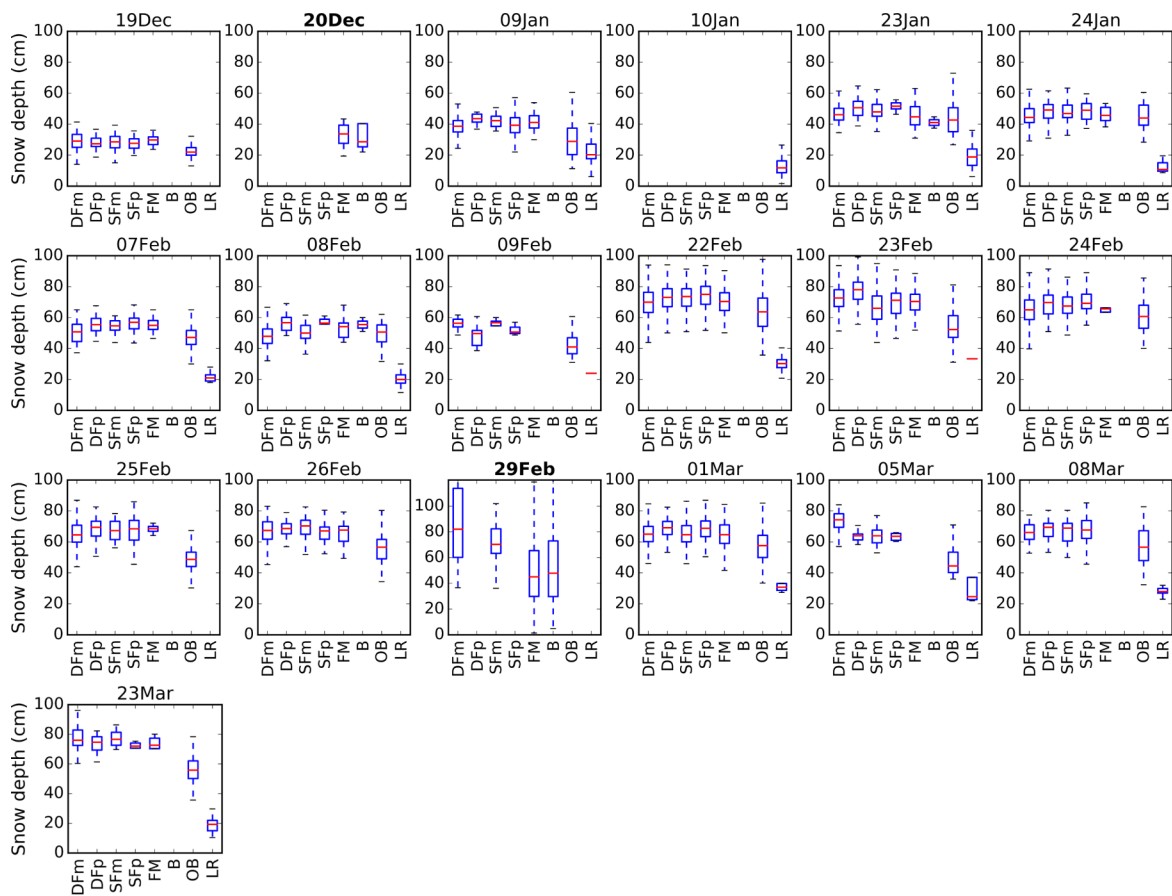

Figure 3: The boxplots of measured snow depth within each land cover group during the different
field measurement days. Measurement days conducted in the Saariselkä test site are indicated in
**bold**.





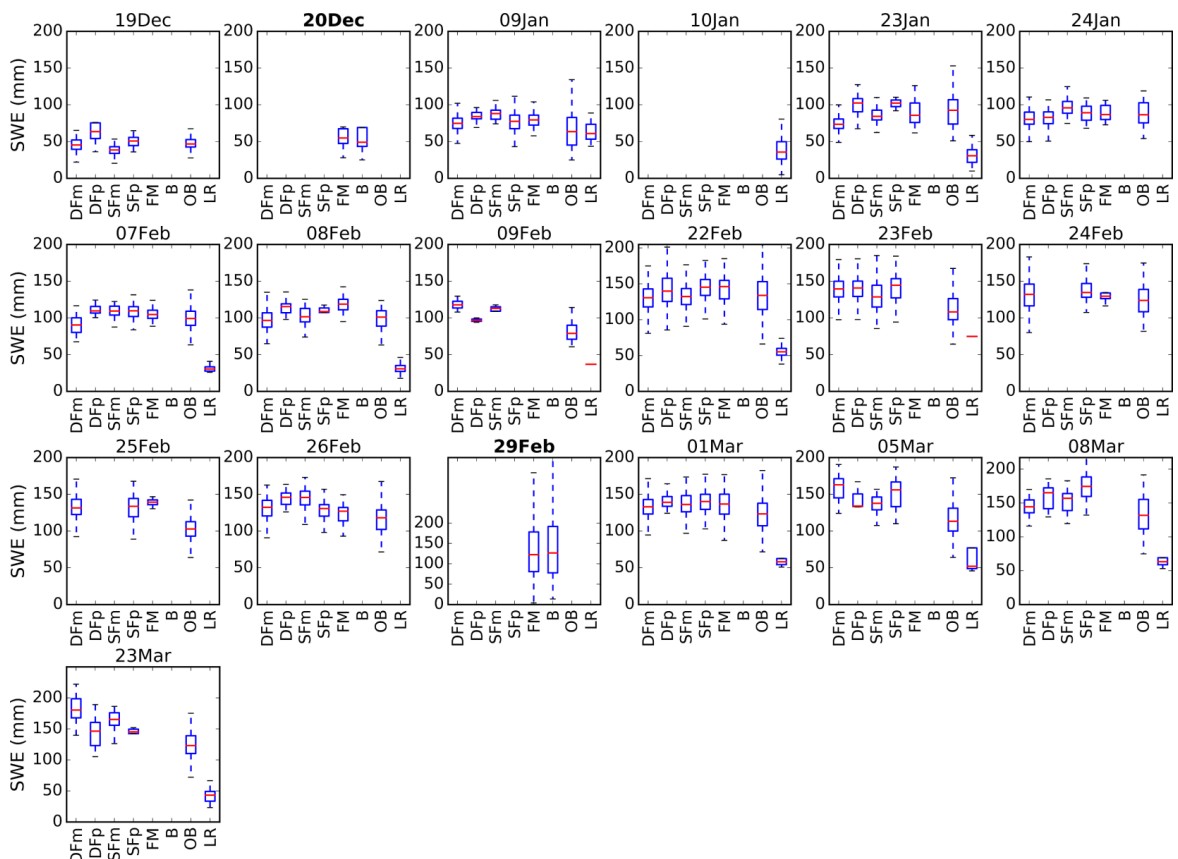

Figure 4: The boxplots of measured SWE within each land cover group during the different field measurement days. Measurement days conducted in the Saariselkä test site are indicated in **bold**.



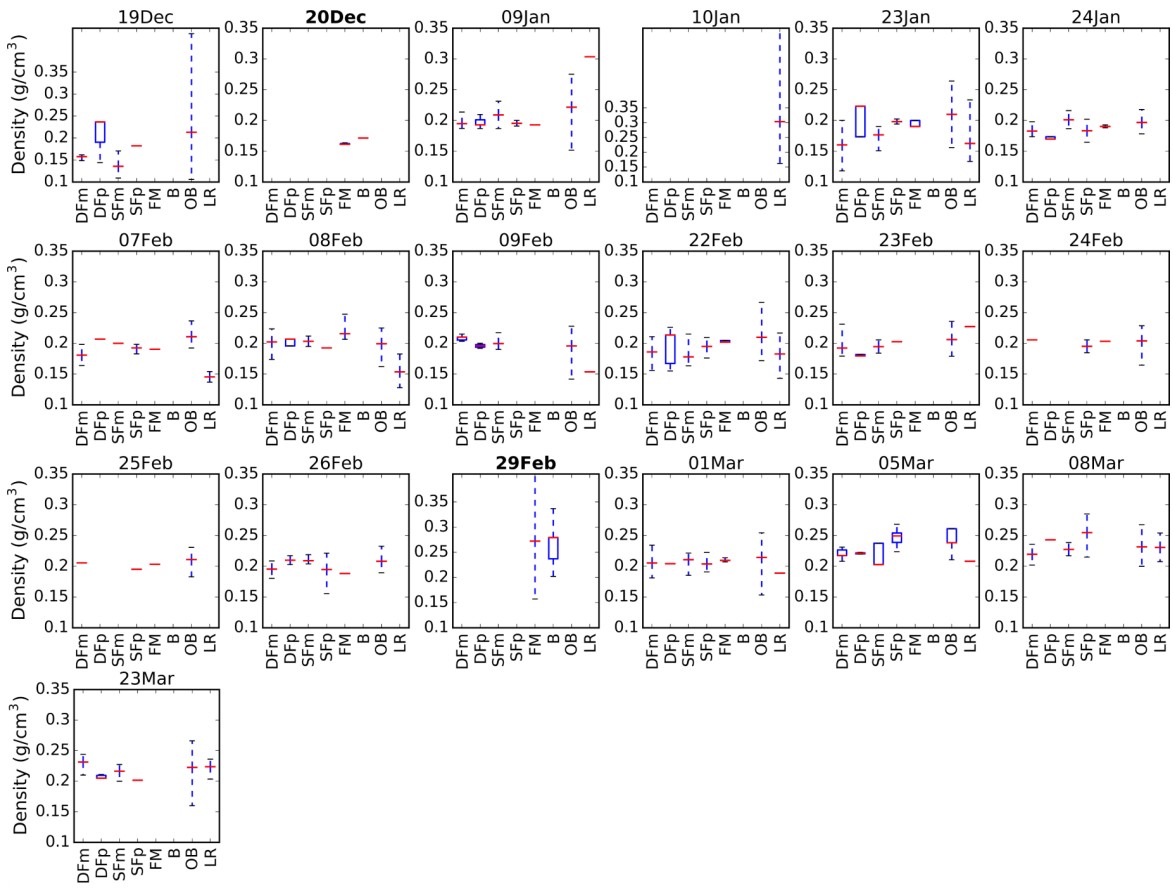

Figure 5: The boxplots of measured snow density within each land cover group during the different
field measurement days. Measurement days conducted in the Saariselkä test site are indicated in
**bold**.

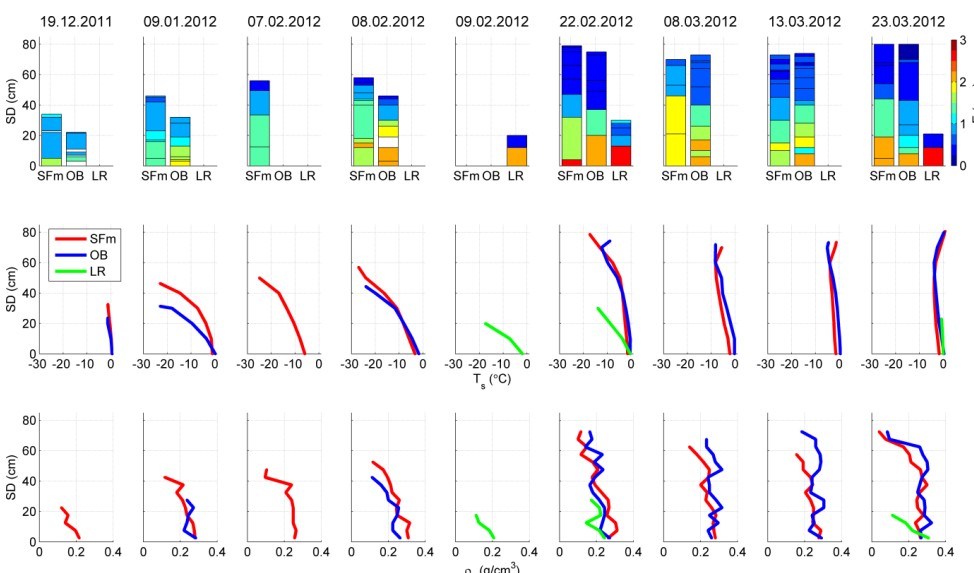

Figure 6: The description of snow stratigraphy during the measurement campaign in three different land cover types; sparse forest on mineral soil (SFm), open bogs (OB), and lake ice (LR). Upper: Bar charts characterize detected snow layers, the maximum diameter of a typical snow grain (E) within each snow layer is indicated with colour. White indicates ice layers where individual snow grains were not detected. Middle: snow temperature profiles. Lower: snow density profiles.



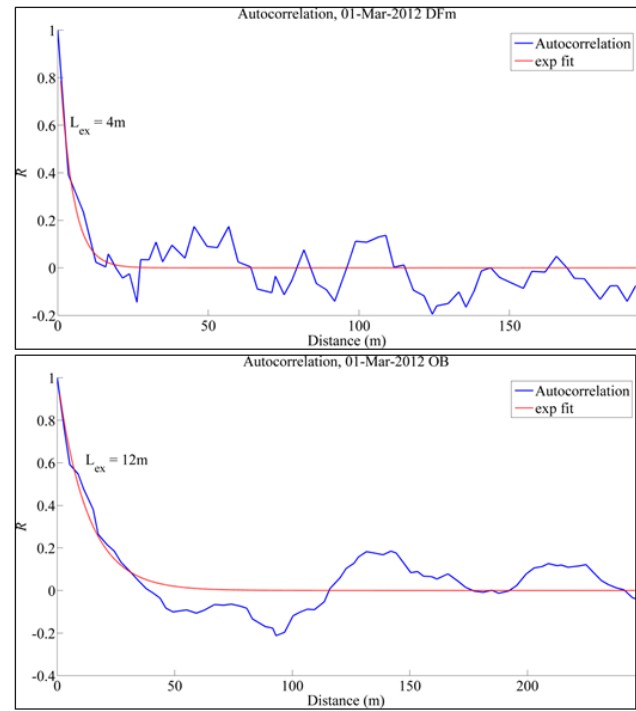

593    Figure 7: Typical exponential fits to the calculated autocorrelation lengths of snow depth measured

594    over forests (DFm, top) and wetlands (OB, bottom) on 1Mar 2012.