# Peer review of "Spatial and Temporal Variation of Bulk Snow Properties in"

_Geoscientific Instrumentation, Methods and Data Systems, 2015_

## Referee Comment (RC1) · Anonymous Referee #1 · 16 Feb 2016

In this paper the authors describe an intensive in situ measurement program over different land cover types in northern Finland during winter 2011-2012. The measurement campaign was carried out over 5 km transects and included manual (every ∼100 m) and Magnaprobe (every ∼2-10 m) measurements of snow depth (SD), along with manual snow corer measurements (every ∼500 m) of snow water equivalent (SWE). Ancillary information was also collected on snowpack structure. The purpose of this detailed measurement campaign was to provide ground-truth for evaluating ESA Snow SAR airborne acquisitions. The paper describes the data collection process, analyzes the spatial and temporal variability in snow cover across nine different land cover types, and makes some conclusions about "optimal" sampling strategies for SD measure-

ments based on the spatial autocorrelation structure. There is no doubt the authors have collected a valuable dataset. However, the paper is a frustrating read because key concepts (e.g. spatial variability of snow cover) were not discussed at the outset, and the methodology evolves throughout the paper instead of being clearly defined at the outset and linked to specific problems/hypotheses. It is also unclear what new findings are being presented and how the study conclusions relate to previously published work. The authors also make a number of sweeping conclusions about "true" snow cover and "optimal" sampling that are (1) based on a limited sample size, (2) do not take into account the error from the fitted spatial model, and (3) do not take into consideration the spatial scales and uncertainty requirements of users. The paper as it stands requires extensive revisions. However, one option the authors might consider is to remove the spatial analysis component of the paper, and resubmit a much shorter paper that describes the dataset and its importance for the snow research community.

Detailed comments:

1. Introduction: the first two paragraphs are peripheral to the study. The focus of the paper is on measurement uncertainties and scaling issues so you need to plunge into this at the outset. The work of Pomeroy and Gray (1995), the seminal paper by Blöschl (1999), the review paper by Clark et al (2011) and more recent work by McCreight et al (2014), and Trujillo and Lehning (2015) should be consulted to help frame the discussion and framing of the problems being addressed in this paper. You should also look at some of the recent papers appearing in the literature looking at detailed spatial variability in snow cover from airborne or UAS lidar (e.g. Zheng et al. 2016). Some discussion of user needs would also be appropriate in the introduction. For some applications such a runoff monitoring over large basins in non-mountainous terrain, spatially averaged SWE information at 10-25 km scale is probably more than adequate when combined with higher resolution satellite snow cover information in a hydrological modelling framework e.g. Bergeron et al (2014).

2. The terms "optimal" and "true snow conditions" are introduced in lines 81-83 without

rigorous definitions or any discussion. In practice, both these terms depend on user requirements.

3. Study objectives (last para page 4): Given that the objectives listed here have been previously studied by a number of investigators, what is unique about the data collected and the proposed data analysis methods that merit publication in GI? The "aims and scope" of GI on the GI homepage may be helpful in responding to this comment.

4. Data and methods: A figure/schematic showing the different measurement methods and their approximate spatial scales would be helpful background information for this section.

5. Second sentence in Section 2.3.1 is difficult to follow. Suggest rewording to "Land cover class was determined based on the GPS coordinates..."

6. Line 149 page 7: Where does the 30% threshold come from?

7. Line 159 page 7: "... has larger effects on the RS ..."

8. Lines 188-195 page 8: Please provide the equation, the definition and some discussion of the correlation length as this is a central part of your analysis method. Since this is obtained from curve fitting, the regression error should also be discussed and presented. It is not entirely clear to me how this statistical property translates to "optimal" sampling e.g. one could fit an autocorrelation function to SWE data collected over 10 or 25 km grid cells and obtain a correlation length corresponding to this scale of information. You don't discuss how rmsd varies with distance but it seems to me this is more important for uncertainty analysis than the spatial autocorrelation i.e. the rmsd may be within operational requirements over a longer distance than suggested by the correlation length. What about interannual and site variability in the length scale? Do you get similar results repeating the measurements in another year and at another location?

9. Line 219 page 9: what does "percentual" mean?
10. Line 239 page 10: Is the "deviation of snow depth" the standard deviation? I suggest you use consistent terminology to avoid confusion.

11. Section 3.3 lines 328-333 talks about results but does not give any! The presentation of analysis results throughout the paper needs to be more focussed and concise.

12. Line 389: Is "measurement frequency" the correct term here?

13. Line 398: I take issue with your conclusion that observing at resolutions higher than Lex does not provide "meaningful statistical information". The relevance of spatial scale depends on the application and the scales of the processes contributing to variability in the snow or snow-related property of interest. For runoff monitoring, synoptic scale events are important for accumulation and melt and these operate at scales much larger than 5 m! Sub-grid scale variability can also be estimated through distributed snow modelling.

14. Line 438: What do you mean by "correct" RS information retrieval? This is subjective terminology.

15. Line 448: The same comment applies to the "true variation of SD" which is a statistical concept. I suggest you revise this sentence to read "... to capture the spatial variation in SD typical of these environments".

16. Lines 449-453: see previous comment in #13. Taking your point to its logical conclusion we should scrap satellites and invest in an army of Lidar-equipped drones for monitoring snow depth :0)

17. Where is the dataset published? I assumed a journal dedicated to datasets would require the dataset to be published online.

References cited:

Bergeron, J., Royer, A., Turcotte, R. and Roy, A. (2014), Snow cover estimation using blended MODIS and AMSR-E data for improved watershed-scale spring streamflow simulation in Quebec, Canada. Hydrol. Process., 28: 4626–4639. doi: 10.1002/hyp.10123

Blöschl, G., 1999. Scaling issues in snow hydrology. Hydrological processes, 13(1415), pp.2149-2175.

Clark, M. P. and Coauthors, 2011: Representing spatial variability of snow water equivalent in hydrologic and land-surface models: A review. Water Resour. Res., 47, doi:10.1029/2011WR010745. http://www.agu.org/pubs/crossref/2011/2011WR010745.shtml (Accessed March 10, 2012).

McCreight, J. L., Slater, A. G., Marshall, H. P., and Rajagopalan, B., 2014: Inference and uncertainty of snow depth spatial distribution at the kilometre scale in the Colorado Rocky Mountains: the effects of sample size, random sampling, predictor quality, and validation procedures, Hydrol. Process., 28, 933–957.

Pomeroy, J.W., and D.M. Gray, 1995: Snowcover - Accumulation, Relocation and Management, National Hydrology Research Institute Science Report No. 7, Saskatoon, Canada, 144 pp.

Trujillo, E. and Lehning, M., 2015: Theoretical analysis of errors when estimating snow distribution through point measurements, The Cryosphere, 9, 1249-1264, doi:10.5194/tc-9-1249-2015.

Zheng et al. 2016: Topographic and vegetation effects on snow accumulation in the southern Sierra Nevada: a statistical summary from lidar data. The Cryosphere, 10, 257-269, 2016 http://www.the-cryosphere.net/10/257/2016/ doi:10.5194/tc-10-257-2016

---

## Referee Comment (RC2) · Anonymous Referee #2 · 24 Feb 2016

**Comments on "Spatial and Temporal Variation of Bulk Snow Properties in North Boreal and Tundra Environments Based on Extensive Field Measurements" by H.-R. Hannula et al., Geosci. Instrum. Method. Data Syst. Discuss.**

**General comments**

This study presents en extensive dataset of snow depth, density and SWE in Lapland, Finland. Different type of land (open area, forest, ...) are studied. The main goal is to provide ground measurements to support remote sensing snow measurements (comparison of data, parameterization, model validation, etc). Using statistical tools, the authors analyze the spatial variability of the above properties and the "optimal" sampling frequency.

The dataset is remarkable, result from a rigorous measurements protocol, and can be useful for different applications. It represents a considerable work that has to be acknowledged.

The spatial variability study is interesting but the novelty of the results should be presented more explicitly. Discussion about previous studies and results comparison are missing. There are more suitable instruments, such as the snow micro pen (Proksch et al 2015), and methods (e.g. Reuter et al. 2015) to study the snow spatial variability (but probably not available during your 2011/2012 campaign). I think it would be helpful for the reader to better express the motivation of this spatial variability study in the context of the FMI work.

The paper is in overall well written. It can be shortened by being more concise in some paragraph of the method and results section. The figures are suitable, except Fig 2, which is not described at all in the paper. The number of table might be reduced by merging some of them together.

**Specific comments**

Line 30: parametrisize → parameterize

Section 2: The organization could be improved for a better clarity.
I suggest
**2.1 Study sites**
      2.1.1 Sodankyla
      2.1.2 Saariselka
      2.1.3 Land cover: you can move here lines 142 to 162.
**2.2 In situ measurements**
Here in my view, you have to organize better and be more concise about your measurements. For instance, I don't see the reason why lines 117-129 and lines 130-138 are 2 paragraphs?
It would be clearer, if it is possible, to divide into subsection such as:
2.2.1 Snow depth & SWE
2.2.2 Snow density:  I will move the density paragraph here
2.2.3 Snow pit measurements
**2.3 Analysis of snow spatial variability**

Line 86: which instruments were set up for the airborne acquisition? Which data did you get from these flights? In my view we need this information to understand your motivation to perform ground measurements of density, SWE…

Line 94: Fig.1 left: you should introduce the abbreviation used for the land cover classes before, otherwise we don't understand the color scale.

Line 107, You should not start your paragraph with details about the airborne, but with the general idea of this paragraph, i.e. line 117 "Manual sampling of SD and SWE along flight transects formed the core of the in situ data collection in support of each SnowSAR acquisition."

Line 107 to 113: I wander if we really need the detailed planning of the airborne acquisitions. I will rather just say that ground measurements were performed during the day of the airborne flights to allow comparison (or the day after, if no strong changes, as mentioned line 114).

Line 124: the description of the tool used for snow depth measurements is missing here. Since you also used the MagnaProbe, you should give a name for this method (latter on, line 199, you define it as conventional method).

Line 127: is there a reference paper on the MagnaProbe? If yes, please cite.

Line 125: the description of the tool used for the SWE measurements is missing.

Line 132: "The main objective…" either this sentence is redundant with line 117, either it should be moved in the beginning of the paragraph.

Line 135: "Additionally, the manual snow measurement…": you have to be more specific, what measurements did you do? Stratigraphy, Ramsonde, temperature profile, etc, with which tools?

Line 163 to 176: I will move this density section above, as suggested in the plan above.

Line 163: Give the equation to retrieve density from depth and SWE, also it is trivial.

Line 171: From your computation of density, is that right that you compute the SWE from depth values in a 10 m radius location? Please clarify in the text. Did you study the error that you do by using 1 density value for all the SWE estimations over a 10 m radius area?

Line 172: "For some data points no density…" either you explained why, either this sentence is not useful since we don't get anything from it.

Line 177-179: should be move in the "snow pit" paragraph.

Line 189: Explanations and references are clearly missing about the autocorrelation method.

Line 199: the information of the snow depth measurements should be moved to the data collection section.

Line 205-214: it might be helpful to give references for all the statistical tests that you are using.

Line 216: delete comma after "investigated".

Throughout the results section: You should give number instead of only use qualitative words like "low values", "higher than". It will give more "dynamic ton" to your writings. I also think that you can be more concise and only mentioned the results that lead to interesting interpretation.
The section could be rename "Results and Interpretation" since you already interprets your results for some of them.

Line 238: it would be nicer to start by an introductive sentence about Figure 3. Explain what is the red line and blue box. Give to the reader a big picture / overview of your SD values, before to go in the details.

Line 240: "generally lower than", "than" is missing

Line 265: I think you can insert a paragraph break before "Regarding…"

Line 291-293: Give values instead of colors.

Line 300: "Temperature profiles reflect the fact that air temperature was the same at all pits measured on the same day; the differences in the snow surface temperatures can be explained by the differing measurement times." How can you distinguish the air temperature from the snow surface temperature in Fig 6?

Line 305 to 311: All this paragraph should be move to the method section when you explain the autocorrelation analysis.

Line 323: What is the mean coefficient of variation?

Line 323: I will write "coefficient of variation" instead of using the abbreviation "CV". A page latter, I would guess that readers would have already forgot this definition.

Line 328 and Table 5: you have to explain the analysis you did in the method section and define all the term "Df", "t-statistic", "p", etc. What does it mean and represent?

Table 5: The term "snow depth" does not appear here and neither in the legend! Please give also the unit.

Line 351: "land cover specific density values": what do you mean? This appellation appears here but was not define before.

Line 390: What do you mean by "true variance"?

Discussion section: In overall, I found very few comparison / discussion of your results with previous studies. In particular, you should point out more clearly what are the new results from your work concerning spatial variability and sampling frequency.

Table 6: The column should be aligned.

Table 3 and 4: Can you merge these two tables together? Ideally 1 or 2 large tables regrouping all the small ones would be even better and make the information easy to find.

Table 3: Please add the unit.

Fig 2: This figure is not commented in the paper, so either there is something interesting about it and you should describe it, either you delete it.

**REFERENCE**
Proksch, M., Löwe, H., & Schneebeli, M. (2015). Density, specific surface area, and correlation length of snow measured by high‐resolution penetrometry. Journal of Geophysical Research: Earth Surface, 120(2), 346-362.
Reuter, B., Richter, B., & Schweizer, J. (2015). Snow instability patterns at the scale of a small basin. Journal of Geophysical Research: Earth Surface.

---

## Referee Comment (RC3) · Anonymous Referee #3 · 3 Mar 2016

General comments:

The paper presents an interesting dataset used for validating remote sensing products; but, at the same time, it is used for analysing spatial distribution of snow characteristics in different landscapes and deriving information that may result useful for considering when planifying snow sampling strategies, specially in such high latitude environment. The paper is quite clear and the main comments I have is about the simplification of applying often a single value of snow density to estimate SWE over a large area. The error introduced by this simplification might be partially quantified using the available landscape units with more than one density measurements. In addition, some statements are supported by references that are not the most commonly used in literature
Interactive
comment

and should be reconsidered, and I miss some other reference that may be useful for the disscusion. In general, the presentation of results and discussion is often mixed. I would suggest to use the "results section" for presenting only the results, and to provide the potential hyphotesis for explaining them in the discussion section. Discussion does not provide any reference, so results are not contrasted with previous research on this topic. In my opinion this question is basic, and authors should modify this section accordingly.

Specific comments

Line 29: Accurate snow: please check font size.

Line 54-57: Probably it is possible to simplify the sentence

Line 79: I see more logical to say snow depth, snow density and snow water equivalent.

Line 134: 22100 measurements

Line 155. I think these references are not the most representatives about the role of canopy density on snow distribution available in international literature. I would reconsider to use more cited and relevant ones.

Line 165: "If more than one SWE points were measured within the same land cover group during the same day, an average of these measurements was used." I thin that if more than one swe data is available for one land class, they should be also used to assess the uncertainty of applying such simplification. For example, it can be shown the differences of density observed in land classes were 2 measurements are, or if there are more density measurements, the difference between each measurement with the average of the other density values available for that land class. Which error may induce this simplification in SWE estimations?

Line 196: I think that the use of sampling frequency is a bit confusing for the readers as is unclear if it refers to time or space. Sampling spacing could be more clear. 239. should be "snow depth on the lakes and rivers"?

[Figure]

Line 272: "this is explained by..." this should be moved to discussion

Line 284. At some point, authors relate the soil characteristics (mineral or organic) with snow density; what does support this assumption?

Line 290: The explanation of figure 6 is rather poor and mostly based in hypothesis, I recommend to go deepen in the explanation of the figure or remove it from the manuscript, as probably it is not very related with the man aim of the manuscript.

Figure 7, the break point to determine Lex might be marked in the figure.

I think that the following references may be considered to discuss the results: Sturm et al., 2010. J. Hydrometeorology: Density and SWE variability in different landscape classes and the impact of errors in density estimation on SWE estimations. Trujillo and Lehning (2015), The Cryosphere: Impact of number of measurements and sampling strategies on estimating snow in profiles or plots of different lengths. López-Moreno et al., 2013. Advances in Water Resources: Spatial variability of the snow and the difficulties to distribute spatially punctual observations.
* * *

---

## Author Comment (AC1) · 12 Apr 2016

General comments:

In this paper the authors describe an intensive in situ measurement program over different land cover types in northern Finland during winter 2011-2012. The measurement campaign was carried out over 5 km transects and included manual (every ∼100 m) and Magnaprobe (every ∼2-10 m) measurements of snow depth (SD), along with manual snow corer measurements (every ∼500 m) of snow water equivalent (SWE). Ancillary information was also collected on snowpack structure. The purpose of this detailed measurement campaign was to provide ground-truth for evaluating ESA Snow SAR airborne acquisitions. The paper describes the data collection process, analyzes

the spatial and temporal variability in snow cover across nine different land cover types, and makes some coclusions about "optimal" sampling strategies for SD measurements based on the spatial autocorrelation structure. There is no doubt the authors have collected a valuable dataset. However, the paper is frustrating read because key concepts (e.g. spatial variability of snow cover) were not discussed at the outset, and the methodology evolves throughout the paper instead of being clearly defined at the outset and linked to specific problems / hypotheses. It is also unclear what new findings are being presented and how the study conclusions relate to previously published work. The authors also make a number of sweeping conclusions about "true" snow cover and "optimal" sampling that are (1) based on a limited sample size, (2) do not take into account the error from the fitted spatial model, and (3) do not take into consideration the spatial scales and uncerainty requirements of users. The paper as it stands requires extensive revisions. However, one option the authors might consider is to remove the spatial analysis component of the paper, and resubmit a much shorter paper that describes the dataset and its importance for the snow research community.

Thank you for the detailed review of the paper. All comments will be now answered. Each comment is followed by the response. The revised version of the paper will be submitted after the author's response for the referee comments have been submitted. We agree that the analysis of the snow spatial variability has been left narrow and the links with previous studies are largely missing. The purpose of the scale analysis was supposed to be made in the special context of the SnowSAR-2 airborne campaign (Di Leo et al., 2015) and this has not been clearly expressed. As suggested by the Referee #1 we will expand the presentation of the data itself and we will compress the scale analysis, retaining only the most significant findings describing the collected snow dataset. The significance of these findings will be explained more elaborately, following suggestions made by the reviewer and will be better explained in the context of the FMI work.

Detailed comments:

[Figure]

1. Introduction: the first two paragraphs are peripheral to the study. The focus of the paper is on measurement uncertainties and scaling issues so you need to plunge into this at the outset. The work of Pomeroy and Gray (1995), the seminal paper by Blöschl (1999), the review paper by Clark et al (2011) and more recent work by McCreight et al (2014), and Trujillo and Lehning (2015) should be consulted to help frame the discussion and framing of the problems being addressed in this paper. You should also look at some of the recent papers appearing in the literature looking at detailed spatial variability in snow cover from airborne or UAS lidar (e.g. Zheng et al. 2016). Some discussion of user needs would also be appropriate in the introduction. For some applications such a runoff monitoring over large basins in non-mountainous terrain, spatially averaged SWE information at 10-25 km scale is probably more than adequate when combined with higher resolution satellite snow cover information in a hydrological modelling framework e.g. Bergeron et al (2014).

The introduction will be rewritten focusing better on the uncertainties and scaling issues as suggested by the Referee#1. The general discussion about the importance of the snow cover information, which is mentioned not to be relevant for the study, will be compressed. The discussion and study objectives will be framed in the context of the ESA SnowSAR-2 airborne acquisitions (Di Leo et al., 2015) with a focus on data description. The in situ dataset was collected to provide ground-truth for the airborne acquisitions, currently offering a 2 m data resolution. However, the snow retrievals (e.g. SWE) sought by the SnowSAR data will not be produced at the native 2m resolution but aggregated up to e.g. 200-500 m. Hence, the analysis of variability of snow properties at scales from several meters and beyond becomes necessary. Keeping this in mind, some discussion about the user needs and scale issues, including the definition of 'scale' and related concepts, in the context of this study and in general, will be added.

2. The terms "optimal" and "true snow conditions" are introduced in lines 81-83 without rigorous definitions or any discussion. In practice, both these terms depend on user requirements.

The "true snow conditions" will be defined as the snow conditions, having distinct pattern of variance and characteristics, at the time of the measurements of this study. However, our only knowledge of these "true" snow conditions is the collected in situ dataset described in this paper, which itself already includes some errors of estimation. As such, the term "apparent snow conditions" suggested and used by Blöschl (1999) will be added to refer to the snow characteristics captured by the in situ dataset. The "optimal" sampling frequency will be defined as a sampling strategy which will capture the variation described by the dataset without significant over- or under-sampling, and which will fulfil the user requirements for the ground-truth data collection for the specific aerial dataset. A rigorous definition of all the terms used, as well as, discussion about the terminology, scale issues, uncertainty issues, and their dependence on the user needs will be added.

3. Study objectives (last para page 4): Given that the objectives listed here have been previously studied by a number of investigators, what is unique about the data collected and the proposed data analysis methods that merit publication in GI? The "aims and scope" of GI on the GI homepage may be helpful in responding to this comment.

The snow in situ measurements described in this study comprise an exceptionally large manually collected dataset of snow conditions in taiga and tundra environments. As mentioned in the review paper of Clark et al. (2011), also recommended by the referee #1, many of the earlier studies with a same count of individual measurements have been automatically collected by using e.g. ground penetrating radar or light detection and ranging (LIDAR) instrument. Also, previous intensive measurement campaigns in northern Scandinavia in a similar kind of environments are limited. As such, the collected dataset is a valuable addition for the snow research community and merits a publication in GI, as the journal features advances in "major national and international field campaigns and observational research programs". In addition to this, the aims of "uncertainty in measurements" and "calibration and data quality assessment" will be touched, although, will not be the main focus in the revised version of the paper.

4. Data and methods: A figure/schematic showing the different measurement methods and their approximate spatial scales would be helpful background information for this section.

A schematic/table to collectively represent all the different measurement methods and their approximate scales will be added.

5. Second sentence in Section 2.3.1 is difficult to follow. Suggest rewording to "Land cover class was determined based on the GPS coordinates..."

The sentence will be reworded as suggested.

6. Line 149 page 7: Where does the 30% threshold come from?

The 30 % threshold value comes from the Corine2012 (Coordinate information on the environment 2012) land cover dataset which was used to determine a generalized land cover class for each measurement GPS point. In Corine2012 dataset, areas with tree cover density exceeding 30 % are classified as dense forests and areas with tree cover density between 10-30 % are classified as sparse forests. The 10 % and 30 % threshold values follow the definitions of the European Environment Agency (EEA) for a 'forest', and for a 'dense forest', respectively. Information of the EEA forest classification and the technical details of the Corine2012 land cover project, being currently part of the Copernicus/GIO land program, can be found from:

http://www.eea.europa.eu/soer-2015/europe/forests

http://land.copernicus.eu/user-corner/technical-library

7. Line 159 page 7: ". . . has larger effects on the RS. . ."

This will be corrected.

8. Lines 188-195 page 8: Please provide the equation, the definition and some discussion of the correlation length as this is a central part of your analysis method. Since this is obtained from curve fitting, the regression error should also be discussed and

presented. It is not entirely clear to me how this statistical property translates to "optimal" sampling e.g. one could fit an autocorrelation function to SWE data collected over 10 or 25 km grid cells and obtain a correlation length corresponding to this scale of information. You don't discuss how rmsd varies with distance but it seems to me this is more important for uncertainty analysis than the spatial autocorrelation i.e. the rmsd may be within operational requirements over a longer distance than suggested by the correlation length. What about interannual and site variability in the length scale? Do you get similar results repeating the measurements in another year and at another location?

The correlation length analysis was done at this scale due to the connection with the SnowSAR-2 campaign. We agree this was not clear and have now emphasized this. The main question we wanted to answer by analysing the correlation length from Magnaprobe measurements (done every ~1-3 m), was to try to estimate how much information will be lost by sampling snow depth with increasing spatial distance (e.g. the original sampling strategy was to measure snow depth only every 100 meters). However, the reviewer is correct that the correlation length alone does not provide a satisfactory answer in this regard. As suggested, we will add an analysis of RMSD with distance.

The equation, the definition, and some discussion of the correlation length, as well as, regression error will be added. As the referee #1 points out, the method or the results are not self-explanatory. The possible variation of the results during different years and different sites will be discussed; however, similar data from other sites or seasons is not available to the authors. It is expected that the absolute values for correlation length vary year and site to site, but the relational differences, e.g. between open and forested areas may hold. In respect of "optimal" sampling, discussion will come back to the definitions also touched in the comment #2.

9. Line 219 page 9: what does "percentual" mean?

The "percentual" refers to %-coverage of each land cover group within the specified 7 km x 10 km area. The word will be replaced with "%-coverage".

10. Line 239 page 10: Is the "deviation of snow depth" the standard deviation? I suggest you use consistent terminology to avoid confusion.

"Standard deviation" will be used instead of just "deviation" throughout the paper.

11. Section 3.3 lines 328-333 talks about results but does not give any! The presentation of analysis results throughout the paper needs to be more focussed and concise.

The actual values of the results will be added. In addition, the chapters for the results and the discussion will be reordered as other anonymous referees commented these sections to be interchangeable.

12. Line 389: Is "measurement frequency" the correct term here?

The term "measurement frequency" will be replaced with "measurement spacing" following the definition by Blöschl (1999) where he suggests that measurement scale consists of three parts: spacing, extent, and support.

13. Line 398: I take issue with your conclusion that observing at resolutions higher than Lex does not provide "meaningful statistical information". The relevance of spatial scale depends on the application and the scales of the processes contributing to variability in the snow or snow-related property of interest. For runoff monitoring, synoptic scale events are important for accumulation and melt and these operate at scales much larger than 5 m! Sub-grid scale variability can also be estimated through distributed snow modelling.

This is right and a more detailed discussion about the scale issues will be added. The results will be discussed specifically in the context of this measurement campaign and its application needs. In addition, it will be discussed, what snow processes might be captured on this scale, and which processes probably work on different scales and would thus, require/be satisfied with denser/sparser measurement spacing.

[Figure]

14. Line 438: What do you mean by "correct" RS information retrieval? This is subjective terminology.

The term of "correct RS information" was meant to refer to the actual snow conditions without the error introduced by the retrieval methods (here, remote sensing). However, as discussed earlier, as the "true" depends on user needs and as such is not good terminology, the sentence will be removed.

15. Line 448: The same comment applies to the "true variation of SD" which is a statistical concept. I suggest you revise this sentence to read "... to capture the spatial variation in SD typical of these environments".

The sentence will be revised as suggested.

16. Lines 449-453: see previous comment in #13. Taking your point to its logical conclusion we should scrap satellites and invest in an army of Lidar-equipped drones for monitoring snow depth :0)

As mentioned previously, the concept of spatial scale will be discussed and the results will be analysed in relation to the user needs of the later SnowSAR aerial data analysis. Indeed, no conclusions, without defining the appropriate scale for the application, can be made.

17. Where is the dataset published? I assumed a journal dedicated to datasets would require the dataset to be published online.

The dataset is now available via the ESA Campaign web page upon registration: https://earth.esa.int/web/guest/campaigns. This information will be updated in the manuscript.

References cited:

Bergeron, J., Royer, A., Turcotte, R. and Roy, A. (2014), Snow cover estimation using blended MODIS and AMSR-E data for improved watershed-scale spring

stream-flow simulation in Quebec, Canada. Hydrol. Process., 28: 4626-4639. Doi:10.1002/hyp.10123

Blöschl, G., 1999: Scaling issues in snow hydrology. Hydrological Processes, 13(1415), pp.2149-2175.

Clark, M. P. and Coauthors, 2011: Representing spatial variability of snow water equivalent in hydrologic and land-surface models: A review. Water Resour. Res., 47, doi:10.1029/2011WR010745.http://www.agu.org/pubs/crossref/2011/2011WR010745.shtml (Accessed March 10, 2012)

McCreight, J.L., Slater, A.G., Marshall, H.P., and Rajagopalan, B., 2014: Inference and uncertainty of snow depth spatial distribution at the kilometre scale in the Colorado Rocky Mountains: the effects of sample size, random sampling, predictor quality, and validation procedures, Hydrol. Process., 28, 933-957.

Pomeroy, J.W., and D.M. Gray, 1995: Snowcover – Accumulation, Relocation and Management, National Hydrology Research Institute Science Report No. 7, Saskatoon, Canada, 144 pp.

Trujillo, E. and Lehning, M., 2015: Theoretical analysis of errors when estimating snow distribution through point measurements, The Cryosphere, 9, 1249-1264,doi:10.5194/tc-9-1249-2015.

Zheng et al. 2016: Topographic and vegetation effects on snow accumulation in the southern Sierra Nevada: a statistical summary from lidar data. The Cryosphere, 10, 257-269, 2016 http://www.the-cryosphere.net/10/257/2016/ doi:10.5194/tc-10-257-2016

References added by the authors:

Di Leo D., Coccia, A., and Meta, A., 2015: Technical Assistance for the Development and Deployment of an X-and Ku-band MiniSAR Airborne System (SnowSAR). ESTEC No. 4000106761-CCN1. (https://earth.esa.int/web/guest/campaigns).

---

## Author Comment (AC2) · 12 Apr 2016

General comments:

This study presents en extensive dataset of snow depth, density and SWE in Lapland, Finland. Different type of land (open area, forest, ...) are studied. The main goal is to provide ground measurements to support remote sensing snow measurements (comparison of data, parameterization, model validation, etc). Using statistical tools, the authors analyze the spatial variability of the above properties and the "optimal" sampling frequency. The dataset is remarkable, result form a rigorous measurements protocol, and can be useful for different applications. It represents a considerable work that has to be acknowledged. The spatial variability study is interesting but the novelty

of the results should be presented more explicitly. Discussion about previous studies and results comparison are missing. There are more suitable instruments, such as the snow micro pen (Proksch et al 2015), and methods (e.g. Reuter et al. 2015) to study the snow spatial variability (but probably not available during your 2011/2012 campaign). I think it would be helpful for the reader to better express the motivation of this spatial variability study in the context of the FMI work. The paper is overall well written. It can be shortened by being more concise in some paragraph of the method and results section. The figures are suitable, except Fig 2, which is not described at all in the paper. The number of table might be reduced by merging some of them together.

Thank you for the detailed review of the paper. All comments will be now answered. Each comment is followed by the response. The revised version of the paper will be submitted after the author's response for the referee comments have been submitted. We agree that the analysis of the snow spatial variability has been left narrow and the links with previous studies are largely missing. The purpose of the scale analysis was supposed to be made in the special context of the SnowSAR-2 airborne campaign (Di Leo et al., 2015) and this has not been clearly expressed. As suggested by the Referee#1 we will expand the presentation of the data itself and we will compress the scale analysis, retaining only the most significant findings describing the collected snow dataset. The significance of these findings will be explained more elaborately, following suggestions made by the reviewer and will be better explained in the context of the FMI work.

Detailed comments:

Line 30: parametrisize → parameterize

Corrected to parameterize

Section 2: The organization could be improved for a better clarity.

I suggest

2.1 Study sites

2.1.1 Sodankylä

2.1.2 Saariselkä

2.1.3 Land cover: you can move here lines 142 to 162.

2.2 In situ measurements

Here in my view, you have to organize better and be more concise about your measurements. For instance, I don't see the reason why lines 117-129 and lines 130-138 are 2 paragraphs? It would be clearer, if it is possible, to divide into subsection such as:

2.2.1 Snow depth & SWE

2.2.2 Snow density: I will move the density paragraph here

2.2.3 Snow pit measurements

2.3 Analysis of snow spatial variability

The section 2 will be reorganized similar or close to as suggested by the referee #2 and more detailed description of the measurement methods and protocols will be added.

Line 86: Which instruments were set up for the airborne acquisition? Which data did you get from these flights? In my view we need this information to understand your motivation to perform ground measurements of density, SWE...

ESA SnowSAR instrument was set up for the airborne acquisition. The instrument is a dual-frequency (9.6 and 17.2 GHz), two-polarization (VV and VH) airborne SAR (synthetic aperture radar) system. The main focus of the aerial campaign was to test and develop an algorithm for SWE ( = snow depth (cm) x snow density (g/cmˆ3)) retrieval. The data provided by SnowSAR are normalized radar cross sections (sigma nought, multi-look) of the target, provided at 2m or alternatively 10 m spatial resolution.

However, the final snow products calculated from the data will be aggregated products with resolutions up to 500m. This provides a motivation to analyse the prevailing snow properties on ground from the meter scale to up to several hundred meters: can the majority of the spatially distributed data, collected e.g. at 100 m intervals, provide an accurate reference for the aggregated SAR products? This will be better explained in the manuscript.

Line 94: Fig.1 left: you should introduce the abbreviation used for the land cover classes before, otherwise we don't understand the color scale.

The abbreviations will be explained more thoroughly and before refering to the Fig.1. This could be done in the section 2.1.3 as suggested by the referee #2.

Line 107: You should not start your paragraph with details about the airborne, but the general idea of this paragraph, i.e. line 117 "Manual sampling of SD and SWE along flight transects formed the core of the in situ data collection in support of each SnowSAR acquisition."

The start of the paragraph will be reorganized as the referee #2 suggests so that the general idea can be understood before going into the details of the flight plan.

Line107-113: I wander if we really need the detailed planning of the airborne acquisitions. I will rather just say that ground measurements were performed during the day of the airborne flights to allow comparison (or the day after, if no strong changes, as mentioned line 114).

The description of the detailed airborne acquisitions can be shortened as it is not the actual focus of this paper. However, generally understanding the motivation and protocol of the in situ measurement collection is seen as an important background.

Line 124: the description of the tool used for snow depth measurements is missing here. Since you also used the MagnaProbe, you should give a name for this method (latter on, line 199, you define it as conventional method).

The description of the snow stake measurements will be added. Conventional method refers to these snow stake measurements which were made less frequently than the Magnaprobe measurements. This will be explained more elaborately in the text, and a description of the tools for all the snow measurements and their approximate scales will be added (as also suggested by the referee #1).

Line 127: Is there a reference paper on the MagnaProbe? If yes, please site.

There is not a reference paper for the Magnaprobe, but the web-page of the manufacturer will be cited: http://www.snowhydro.com/products/column2.html.

125: the description of the tool used for the SWE measurements is missing.

SWE coring tubes manufactured and calibrated by the FMI were used. The description of the tool will be added.

Line 132: "The main objective..." either this sentence is redundant with line 117, either it should be moved in the beginning of the paragraph.

The sentence will be moved to the place where the general idea behind the in situ measurement collection is described (to the beginning of the whole section).

Line 135: "Additionally, the manual snow measurement...": you have to be more specific, what measurements did you do? Stratigraphy, Ramsonde, temperature profile, etc, with which tools?

Here, specifically the snow pit measurements executed on a weekly basis at FMI-ARC are meant. We will add a description what measurements and which tools these snow pit measurements include, as now, only a publication describing the whole measurement procedure is cited.

Line 163-176: I will move this density section above, as suggested in the plan above.

See the answer for the comment about the section 2.

Line 163: Give the equation to retrieve density from depth and SWE, also it is trivial.

The equation will be added.

Line 171: From your computation of density, is that right that you compute the SWE from snow depth values in a 10 m radius location? Please clarify in the text. Did you study the error that you do by using 1 density value for all the SWE estimations over a 10 m radius area?

One density value was averaged within one land cover group, but the snow depth measurements at each point consisted of 3 independent snow stake measurements within a 10 m radius. In case of the Magnaprobe measurements, only one independent measurement at each GPS point was done, and thus, they represent even smaller area. A description of all the measurements and the used datasets and their approximate scales will be added for a clarification. The error introduced by the density averaging was not discussed, but this kind of analysis will be added by comparing the effect of averaging within land cover types, where several density measurement points are available.

Line 172: "For some data points no density..." either you explained why, either this sentence is not useful since we don't get anything from it.

The sentence will be replaced with an explanation about why it was not possible to determine SWE for all the data points where snow depth was measured.

Line 177-179: should be move in the "snow pit" paragraph.

Lines will be moved to the snow pit paragraph as suggested by the referee #2.

Line 189: Explanations and references are clearly missing about the autocorrelation method.

Discussion and references for the autocorrelation method will be added.

Line 199: The information of the snow depth measurements should be moved to the

data collection section.

The snow depth info will be moved to the data collection section as suggested.

Line 205-214: it might be helpful to give references for all the statistical tests that you are using.

The references for the statistical tests will be added. These were originally left out as they were considered as general knowledge.

Line 216: delete comma after "investigated".

Comma will be deleted.

Throughout the results section: You should give number instead of only use qualitative words like "low values", "higher than". It will give more "dynamic ton" to your writings. I also think that you can be more concise and only mentioned the results that lead to interesting interpretation. The section could be rename "Results and Interpretation" since you already interprets your results for some of them.

This will be corrected in the revised version. Either the results and discussion will be clearly separated or then these two sections will be combined as suggested by the referee #2.

Line 238: it would be nicer to start by an introductive sentence about Figure 3. Explain what is the red line and blue box. Give to the reader a big picture / overview of your SD values, before to go in the details.

A general explanation of how the data is described in Figure 3 will be added in the beginning of the section to help the reader.

Line 240: "generally lower than", "than" is missing

'Than' will be added.

Line 265: I think you can insert a paragraph break before "Regarding. . ."

Paragraph break will be added.

Lines 291-293: Give values instead of colors.

Colors will be replaced with actual values.

Line 300: "Temperature profiles reflect the fact that air temperature was the same at all pits measured on the same day; the differences in the snow surface temperatures can be explained by the differing measurement times." How can you distinguish the air temperature from the snow surface temperature in Fig 6?

The first clause of the sentence will be removed and only the snow surface temperatures will be mentioned, as the variance of air temperatures cannot be seen from the referred Fig 6.

Line 305-311: All this paragraph should be move to the method section when you explain the autocorrelation analysis.

We will move these lines to the method section, excluding the last line which is already explaining the results.

Line 323: What is the mean coefficient of variation?

Coefficient of variation is the standard deviation divided by the mean. This was calculated for each of the 8 land cover groups for each measurement day. The results of each land cover group were averaged to represent the whole measurement campaign period, and the mean coefficient of variation is referring to these averaged values of coefficient of variation. A more elaborate explanation will be added to the method section.

Line 323: I will write "coefficient of variation" instead of using the abbreviation "CV". A page latter, I would guess that readers would have already forgot this definition.

Abbreviation "CV" will be removed and 'coefficient of variation' will be used instead.

Line 328 and Table 5: you have to explain the analysis you did in the method section and define all the term "Df", "t-statistic", "p", etc. What does it mean and represent?

An explanation for each statistical test will be added and the terms will be defined. Originally this was left out as the tests were kept as 'basic' knowledge. However, adding the background information helps to understand the results and what they represent. Nonetheless, all the terms should be properly defined.

Table 5: The term "snow depth" does not appear here and neither in the legend! Please give also the unit.

The term "snow depth" and the unit (cm) will be added.

Line 351: "land cover specific density values": what do you mean? This appellation appears here but was not define before.

The averaged density value (averaged from all of the available density measurements within each land cover group during the same measurement day) is meant. The definition will be added to the data and methods section handling the calculation of the density values and a consistent terminology will then be used throughout the paper to refer to these density values. See the answer to comment for Line 163.

Line 390: What do you mean by "true variance"?

The "true variance" is referring to the natural pattern of snow depth variance at the time of the measurements. However, our only knowledge of this variance is the collected in situ dataset described in this paper, which itself already includes some errors of estimation, and as such, does not describe the "true variance". As also noted by the referee #1 this concept have been used without rigorous definition, so we will add definitions for different concepts already in the introduction, mostly following the suggested terminology by Blöschl (1999).

Discussion section: In overall, I found very few comparison / discussion of your results with previous studies. In particular, you should point out more clearly what are the new

results from your work concerning spatial variability and sampling frequency.

We agree that links with previous studies are largely missing and we will correct this in the revised version of the paper. In the revised version we will also concentrate on the description of the dataset and the success of the measurement campaign's sampling strategy to catch the apparent variance in the snow properties during the campaign. As such, the results, which are based on exceptionally large manual survey, largely verify the results of previous studies, and can be used when planning similar kind of campaigns in the same kind of environments. The dataset is interesting for deeper analysis for various applications in the future.

Table 6: The column should be aligned.

The column will be aligned.

Table 3 and 4: Can you merge these two tables together? Ideally 1 or 2 large tables regrouping all the small ones would be even better and make the information easy to find.

Tables 3 and 4, and 2 will be merged to create one new table. This indeed, helps to find and interpret the results when they are next to the explanations of the land cover abbreviations.

Table 3: Please add the unit.

The unit (m) will be added.

Fig 2: This figure is not commented in the paper, so either there is something interesting about it and you should describe it, either you delete it.

The Fig.2 was added to show the weather conditions during and between the measurement days (temperature change, precipitation event, major winds). This was done to justify the use of density information from the previous or the subsequent measurement day in case there was no density information for a specific land cover group during the

measurement day in concern. However, this will be just explained in the text and the Fig. 2 will be removed as it does not add anything interesting to the paper.

References:

Proksch, M., Löwe, H., & Schneebeli, M. (2015). Density, specific surface area, and correlation length of snow measured by high – resolution penetrometry. Journal of Geophysical Research: Earth Surface, 120(2), 346-362.

Reuter, B., Richter, B., & Schweizer, J. (2015). Snow instability patterns at the scale of a small basin. Journal of Geophysical Research: Earth Surface.

References added by the authors:

Blöschl, G. (1999). Scaling issues in snow hydrology. Hydrological Processes, 13, 2149-2175.

Di Leo D., Coccia, A., and Meta, A. (2015). Technical Assistance for the Development and Deployment of an X-and Ku-band MiniSAR Airborne System (SnowSAR). ESTEC No. 4000106761-CCN1. (https://earth.esa.int/web/guest/campaigns).

---

## Author Comment (AC3) · 12 Apr 2016

General comments:

The paper presents an interesting dataset used for validating remote sensing products; but, at the same time, it is used for analysing spatial distribution of snow characteristics in different landscapes and deriving information that may result useful for considering when planifying snow sampling strategies, specially in such high latitude environment. The paper is quite clear and the main comments I have is about the simplification of applying often a single value of snow density to estimate SWE over a large area. The error introduced by this simplification might be partially quantified using the available landscape units with more than one density measurements. In addition, some statements supported by references that are not the most commonly used in literature and should be considered, and I miss some other reference that may be useful for the discussion. In general, the presentation of results and discussion is often mixed. I would suggest to use the "results section" for presenting only the results, and to provide the potential hypotheses for explaining them in the discussion section. Discussion does not provide any reference, so results are not contrasted with previous research on this topic. In my opinion this question is basic, and authors should modify this section accordingly.

Thank you for the detailed review of the paper. All comments will be now answered and each comment is followed by the response. The revised version of the paper will be submitted after the author's response for the referee comments have been submitted. We agree that the analysis of the snow spatial variability has been left narrow and the links with previous studies are largely missing. The purpose of the scale analysis was supposed to be made in the special context of the SnowSAR-2 airborne campaign (Di Leo et al., 2015) and this has not been clearly expressed. As suggested by the Referee#1 we will expand the presentation of the data itself and we will compress the scale analysis, retaining only the most significant findings describing the collected snow dataset. The significance of these findings will be explained more elaborately, following suggestions made by the reviewers and will be better explained in the context of the FMI work. In addition, analysis of the uncertainty, introduced by using only one density value to estimate SWE over large area, will be added.

Detailed comments:

Line 29: Accurate snow: please check font size.

The font size will be corrected.

Line 54-57: Probably it is possible to simplify the sentence.

The sentence will be reworded.

Line 79: I see more logical to say snow depth, snow density and snow water equivalent.

The abbreviation of "SD" will be replaced with "snow depth" throughout the paper. However, using "SWE" makes sentences more compacted, and is generally known and used abbreviation, so we would like to keep that also in the revised version of the paper.

Line 134: 22100 measurements

The space will be removed.

Line 155: I think these references are not the most representatives about the role of canopy density on snow distribution available in international literature. I would reconsider to use more cited and relevant ones.

The international literature representing the role of canopy density on snow distribution will be studied to find more appropriate and more cited studies to refer.

Line 165: "If more than one SWE points were measured within the same land cover group during the same day, an average of these measurements was used." I thin that if more than one swe data is available for one land class, they should be also used to assess the uncertainty of applying such simplification. For example, it can be shown the differences of density observed in land classes were 2 measurements are, or if there are more density measurements, the difference between each measurement with the average of the other density values available for that land class. Which error may induce this simplification in SWE estimations?

The error introduced by the density simplification will be estimated and discussed following the suggestions of the reviewer #3.

Line 196: I think that the use of sampling frequency is a bit confusing for the readers as is unclear if it refers to time or space. Sampling spacing could be more clear.

We will replace "sampling frequency" with "sampling spacing" in the revised version of

the paper.

Line 239: Should be "snow depth on the lakes and rivers"?

"in" will be corrected to "on".

Line 272: "this is explained by. . ." this should be moved to discussion

The sentence will be moved to discussion.

Line 284: At some point, authors relate the soil characteristics (mineral or organic) with snow density; what does support this assumption?

This was related to Fig. 5 where the variation in snow density within each land cover group was presented. It was mentioned that it was difficult to find distinct differences is snow density between different forested land cover groups; often the median of snow density in dense forests on mineral soil was lower than in the other forested land cover groups, but this did not hold throughout the campaign period, as noted on next sentence starting on line 285. We suggest that we remove this part and mention only that no clear difference in snow density between the forested land cover groups was found.

Line 290: The explanation of figure 6 is rather poor and mostly based in hypothesis, I recommend to go deepen in the explanation of the figure or remove it from the manuscript, as probably it is not very related with the man aim of the manuscript.

We will deepen the explanation of the Fig. 6. As in the revised version of the manuscript we will concentrate more on the description of the dataset, and will also concentrate on the compressed scale analysis in the context of this specific measurement campaign, we see these snow pit measurements as a valuable information for later interests of analysis.

Figure 7: the break point to determine Lex might be marked in the figure.

The break point will be added.

References:

Sturm et al., 2010. J. Hydrometeorology: Density and SWE variability in different landscape classes and the impact of errors in density estimation of SWE estimations.

Trujillo and Lehning (2015), The Cryosphere: Impact of number of measurements and sampling strategies on estimating snow in profiles or plots of different lengths.

López-Moreno et al., 2013. Advances in Water Resources: Spatial variability of the snow and the difficulties to distribute spatially punctual observations.

References added by the authors:

Di Leo D., Coccia, A., and Meta, A., 2015: Technical Assistance for the Development and Deployment of an X-and Ku-band MiniSAR Airborne System (SnowSAR). ESTEC No. 4000106761-CCN1. (https://earth.esa.int/web/guest/campaigns).